# MULTIOBJECTIVE STOCHASTIC LINEAR BANDITS UNDER LEXICOGRAPHIC ORDERING

## ABSTRACT

This paper studies the multiobjective stochastic linear bandit (MOSLB) model under lexicographic ordering, where the agent aims to simultaneously maximize $m$ objectives in a hierarchical manner. This model has various real-world scenarios, including water resource planning and radiation treatment for cancer patients. However, there is no effort on the general MOSLB model except a special case called multiobjective multi-armed bandits. Previous literature provided a suboptimal algorithm for this special case, which enjoys a regret bound of $\widetilde{O}(T^{2/3})$ under a priority-based regret measure. In this paper, we propose an algorithm achieving the almost optimal regret bound $\widetilde{O}(d\sqrt{T})$ for the MOSLB model, and its metric is the general regret. Here, $d$ is the dimension of arm vector and $T$ is the time horizon. The major novelties of our algorithm include a new arm filter and a multiple trade-off approach for exploration and exploitation. Experiments confirm the merits of our algorithms and provide compelling evidence to support our analysis.

## 1 INTRODUCTION

Sequential decision-making under uncertainty arises in numerous real-world scenarios, such as medical trials (Robbins, 1952), recommendation systems (Bubeck & Cesa-Bianchi, 2012), and autonomous driving (Huang et al., 2019). This has motivated the development of the stochastic multi-armed bandit (MAB) model, where the agent repeatedly selects an arm from $K$ arms and receives a single-valued reward sampled from a fixed but unknown distribution specific to the selected arm (Agrawal, 1995; Li et al., 2010a; Xue et al., 2020; Ghosh & Sankararaman, 2022). The agent aims to minimize the regret, which is the cumulative difference between the expected reward of the selected arm and that of the best arm. Furthermore, the aforementioned scenarios can be better modeled if multiple objectives are considered. An example is an online advertising system where the agent not only needs to maximize the click-through rate but also the click-conversion rate (Rodriguez et al., 2012). Therefore, a natural extension of MAB is replacing the single-valued reward with a vector, known as multiobjective multi-armed bandits (MOMAB) (Drugan & Nowe, 2013).

A general framework of MOMAB is a $T$-round sequential decision-making system (Drugan & Nowe, 2013), where the agent chooses an arm $a_t$ from the given arm set $\{1, 2, \ldots, K\}$ at the $t$-th round and receives a reward vector $[y^1(a_t), y^2(a_t), \ldots, y^m(a_t)] \in \mathbb{R}^m$ whose $i$-th element is a random variable with expectation $\mathbb{E}[y^i(a_t)] = \mu^i(a_t), i \in \{1, 2, \ldots, m\}$. Most of the existing work evaluates the performance of the agent by Pareto regret (Van Moffaert et al., 2014; Turgay et al., 2018; Lu et al., 2019), which regards all objectives as equivalent and minimizing the regret of any objective can guarantee a sublinear Pareto regret bound (Xu & Klabjan, 2023, Theorem 4.1). Therefore, if the evaluation criterion is Pareto regret, the agent can select any of the $m$ objectives to optimize and ignore other objectives, which is unreasonable.

To deal with this inherent drawback, the lexicographic order is adopted to distinguish the importance among different objectives (Ehrgott, 2005). In this setting, the priority over $m$ objectives is given by indices, such that the $i$-th objective has a higher priority than the $j$-th objective if $i < j$. For the bandit model, given two arms $a$ and $a'$ with expected rewards $\boldsymbol{\mu}(a) = [\mu^1(a), \mu^2(a), \ldots, \mu^m(a)]$ and $\boldsymbol{\mu}(a') = [\mu^1(a'), \mu^2(a'), \ldots, \mu^m(a')]$, arm $a$ is said to **lexicographically dominate** arm $a'$, denoted by $a \succ_{lex} a'$, if and only if $\mu^1(a) > \mu^1(a')$ or there exists some $i^* \in \{2, \ldots, m\}$, such that $\mu^i(a) = \mu^i(a')$ for $1 \leq i \leq i^* - 1$ and $\mu^{i^*}(a) > \mu^{i^*}(a')$. An arm $a_*$ is said to be **lexicographic optimal** if and only if any other arm does not lexicographically dominate it.

Hüyük & Tekin (2021) was the first to explore the MOMAB model under lexicographic ordering and proposed a priority-based regret,

$$\widehat{R}^i(T) = \sum_{t=1}^{T} \left( \mu^i(a_*) - \mu^i(a_t) \right) \mathbb{I} \left( \mu^j(a_*) = \mu^j(a_t), 1 \leq j \leq i-1 \right) \tag{1}$$

where $a_*$ denotes the lexicographic optimal arm and $\mathbb{I}(\cdot)$ is the indicator function. Utilizing this regret, Hüyük & Tekin (2021) developed an algorithm with a regret bound $\widetilde{O}((KT)^{2/3})$, which is suboptimal since the optimal regret bound for existing single objective MAB algorithms is $O(K \log T)$ (Lai & Robbins, 1985). On the other hand, the MOMAB model neglects the contextual information in real-world applications, such as user preferences and news features in news recommendation systems, which could be employed to guide the decision-making process (Li et al., 2010b).

To incorporate contextual information into the decision-making process, a natural approach is to utilize the stochastic linear bandit (SLB) model. The SLB model has been widely researched in the single objective bandit field (Auer, 2002; Dani et al., 2008; Chu et al., 2011; Abbasi-yadkori et al., 2011; Alieva et al., 2021; Zhu & Mineiro, 2022; He et al., 2022; Yang et al., 2022), and here we extend it to multiobjective setting by formalizing the multiobjective stochastic linear bandit (MOSLB) model. In MOSLB, the agent selects an arm $\boldsymbol{x}_t$ from the given arm set $\mathcal{D}_t \subset \mathbb{R}^d$ at the $t$-th round and then receives a stochastic reward vector $[y_t^1, y_t^2, \ldots, y_t^m] \in \mathbb{R}^m$ satisfying

$$\mathbb{E}[y_t^i | \boldsymbol{x}_t, \mathcal{F}_{t-1}] = \langle \boldsymbol{\theta}_*^i, \boldsymbol{x}_t \rangle, i = 1, 2, \ldots, m \tag{2}$$

where $y_t^i$ represents the reward of the $i$-th objective, $\boldsymbol{\theta}_*^i$ denotes the unknown parameters for the $i$-th objective, and $\mathcal{F}_{t-1} = \{\boldsymbol{x}_1, \boldsymbol{x}_2, \ldots, \boldsymbol{x}_{t-1}\} \cup \{y_1^1, y_2^1, \ldots, y_{t-1}^1\} \cup \ldots \cup \{y_1^m, y_2^m, \ldots, y_{t-1}^m\}$ constitutes a $\sigma$-filtration of events up to $t$. Meanwhile, a common assumption on the bandit problem is that the stochastic rewards are sub-Gaussian with a fixed parameter $R \geq 0$, that is, for any $\beta \in \mathbb{R}$,

$$\mathbb{E}[e^{\beta y_t^i} | \boldsymbol{x}_t, \mathcal{F}_{t-1}] \leq \exp \left( \frac{\beta^2 R^2}{2} \right), i = 1, 2, \ldots, m. \tag{3}$$

To evaluate the performance of the agent, we adopt the general regret for single objective SLB (Auer, 2002), such that

$$R^i(T) = \sum_{t=1}^{T} \langle \boldsymbol{\theta}_*^i, \boldsymbol{x}_t^* - \boldsymbol{x}_t \rangle, i = 1, 2, \ldots, m \tag{4}$$

where $\boldsymbol{x}_t^*$ indicates the lexicographic optimal arm in $\mathcal{D}_t$. Clearly, $R^i(T)$ is more stringent than $\widehat{R}^i(T)$ because $\widehat{R}^i(T)$ disregards the regret of $t$-th round when the indicator function is false, whereas $R^i(T)$ accumulates all instantaneous regret.

Existing optimal algorithms for single objective SLB exhibit the regret bound $\widetilde{O}(d\sqrt{T})$ (Dani et al., 2008; Abbasi-yadkori et al., 2011). Therefore, a compelling and non-trivial challenge is to achieve the regret bound $\widetilde{O}(d\sqrt{T})$ for the MOSLB under lexicographic ordering. In line with the standard SLB model (Dani et al., 2008), the sequence of decision sets $\{\mathcal{D}_1, \mathcal{D}_2, \ldots, \mathcal{D}_T\}$ are compact and determined before the game starts. Thus, we **claim** that there exists some $\lambda \geq 0$, the expected rewards of different objectives satisfy

$$\langle \boldsymbol{\theta}_*^i, \boldsymbol{x} - \boldsymbol{x}_t^* \rangle \leq \lambda \cdot \max_{j \in [i-1]} \langle \boldsymbol{\theta}_*^j, \boldsymbol{x}_t^* - \boldsymbol{x} \rangle, \ i = 2, 3, \ldots, m \tag{5}$$

for any $\boldsymbol{x} \in \mathcal{D}_t, t \in [T]$. Appendix A shows our claim is true. We want to emphasize two important properties of the proposed parameter $\lambda$. Firstly, measuring the relative rate at which different objective values change with respect to the decision is sufficient to provide an upper bound for $\lambda$. To illustrate this point, we provide a simple example involving two objectives and a fixed arm set $\mathcal{D}$. For any $\boldsymbol{x}, \boldsymbol{x}' \in \mathcal{D}$, if $|\langle \boldsymbol{\theta}_*^1, \boldsymbol{x} - \boldsymbol{x}' \rangle| \geq L_1$ can guarantee $|\langle \boldsymbol{\theta}_*^2, \boldsymbol{x} - \boldsymbol{x}' \rangle| \leq L_2$, then we have $\lambda \leq L_2/L_1$. $L_2/L_1$ is feasible as different objectives are related to each other in various applications, such as water resource planning (Weber et al., 2002) and radiation treatment for cancer patients (Jee et al., 2007). Secondly, $\lambda$ captures the complexity of identifying the optimal arm $\boldsymbol{x}_t^*$ within $\mathcal{D}_t$. Specifically, if $\lambda$ is exceptionally large, there exists $\boldsymbol{x} \in \mathcal{D}_t$ that yields substantially larger rewards than the optimal arm $\boldsymbol{x}_t^*$ for the $i$-th objective, while maintaining similar rewards for the preceding $i-1$ objectives, making the identification of the optimal arm challenging.

---

[1] For a positive integer $i$, $[i]$ denotes the set $\{1, 2, \ldots, i\}$.

To the best of our knowledge, this paper is the first attempt to investigate the MOSLB model under lexicographic ordering. With the prior knowledge $\lambda$, we develop an algorithm that attains a general regret bound of $\widetilde{O}((\lambda^{i-1} + 1)d\sqrt{T})$ for the $i$-th objective, $i \in [m]$. This bound is almost optimal in terms of $d$ and $T$, as the lower bound for the single objective SLB problem is $\Omega(d\sqrt{T})$ (Dani et al., 2008). Our algorithm improves upon the previous bound $\widetilde{O}((KT)^{2/3})$ in the most recent study of Hüyük & Tekin (2021), which focused on the MOMAB model. Furthermore, we extend the metric of the lexicographically ordered multiobjective bandit problem from the priority-based regret (1) to the general regret (4), which more accurately evaluates the performance of algorithms. The main innovations of our algorithm include a new arm filter and a multiple trade-off approach for exploration and exploitation, which can be easily adapted to other bandit models, such as generalized linear bandits (Jun et al., 2017) and Lipschitz bandits (Bubeck et al., 2011).

## 2  RELATED WORK

In this section, we provide a literature review on stochastic bandits and multiobjective bandits. Throughout the paper, $\|\boldsymbol{x}\|$ is the $\ell_2$-norm of vector $\boldsymbol{x} \in \mathbb{R}^d$. Additionally, the induced norm of $\boldsymbol{x}$ by a positive definite matrix $V \in \mathbb{R}^{d\times d}$ is denoted as $\|\boldsymbol{x}\|_V = \sqrt{\boldsymbol{x}^\top V \boldsymbol{x}}$.

### 2.1  STOCHASTIC BANDITS

The seminal work of Lai & Robbins (1985) not only introduced a stochastic MAB algorithm with a regret bound of $O(K \log T)$ but also established a matching lower bound. Auer (2002) extended the bandit algorithm to the linear model with finite arms and developed the SupLinRel algorithm, which employs a sophisticated device to decouple reward dependence, yielding a regret bound of $\widetilde{O}(\sqrt{dT})$. In the context of infinite-armed stochastic linear bandits, Dani et al. (2008) first applied the confidence region technique to deduce the upper confidence bound for the expected rewards of infinite arms, resulting in a regret bound of $\widetilde{O}(d\sqrt{T})$ that matches the given lower bound $\Omega(d\sqrt{T})$. A subsequent study by Abbasi-yadkori et al. (2011) offered a new analysis for the algorithm of Dani et al. (2008) and enhanced the regret bound by a logarithmic factor.

The Upper Confidence Bound (UCB) framework is a widely-used technique for balancing exploration and exploitation in the decision-making process, which first computes the confidence bound of forthcoming rewards through historical trials and then selects the arm with the highest upper confidence bound (Auer et al., 2002; Abbasi-yadkori et al., 2011; Bubeck et al., 2015; Hu et al., 2021; Li et al., 2022; Masoudian et al., 2022; Feng et al., 2022; Jin et al., 2022). To illustrate the UCB technique utilized in the SLB model, we take the classical algorithm OFUL as an example (Abbasi-yadkori et al., 2011). With trials up to the $t$-th round, OFUL minimizes the square loss of the action-reward pairs $\{(\boldsymbol{x}_1, y_1), (\boldsymbol{x}_2, y_2), \ldots, (\boldsymbol{x}_{t-1}, y_{t-1})\}$ to estimate the inherent parameters $\boldsymbol{\theta}_*$, such that,

$$\hat{\boldsymbol{\theta}}_t = \underset{\boldsymbol{\theta} \in \mathbb{R}^d}{\arg\min} \|X_t \boldsymbol{\theta} - Y_t\|^2 + \|\boldsymbol{\theta}\|^2 \tag{6}$$

where $X_t = [\boldsymbol{x}_1, \boldsymbol{x}_2, \ldots, \boldsymbol{x}_{t-1}] \in \mathbb{R}^{(t-1)\times d}$ is the matrix composed of selected arm vectors, and $Y_t = [y_1, y_2, \ldots, y_{t-1}] \in \mathbb{R}^{(t-1)\times 1}$ is the vector composed of historical rewards. Using the estimator $\hat{\boldsymbol{\theta}}_t$, OFUL constructs a confidence region $\mathcal{C}_t$ where the inherent parameter lies in with high probability, such that

$$\mathcal{C}_t = \{\boldsymbol{\theta} \mid \|\boldsymbol{\theta} - \hat{\boldsymbol{\theta}}_t\|_{V_t} \le \alpha_t\} \tag{7}$$

where $\alpha_t = O(\sqrt{d \log(t)})$ and $V_t = X_t^\top X_t + I_d$. Finally, OFUL selects the most promising arm $\boldsymbol{x}_t$ through bilinear optimization,

$$(\boldsymbol{x}_t, \tilde{\boldsymbol{\theta}}_t) = \underset{x \in \mathcal{D}_t, \boldsymbol{\theta} \in \mathcal{C}_t}{\arg\max} \langle \boldsymbol{x}, \boldsymbol{\theta} \rangle. \tag{8}$$

Considering that the confidence region $\mathcal{C}_t$ is an ellipse, a simple application of the Lagrange method shows that the upper confidence bound for the arm $\boldsymbol{x} \in \mathcal{D}_t$ is

$$u_t(\boldsymbol{x}) = \langle \hat{\boldsymbol{\theta}}_t, \boldsymbol{x} \rangle + \alpha_t \|\boldsymbol{x}\|_{V_t^{-1}}, \tag{9}$$

where $\langle \hat{\boldsymbol{\theta}}_t, \boldsymbol{x} \rangle$ is an unbiased estimation of $\langle \boldsymbol{\theta}_*, \boldsymbol{x} \rangle$, and $\alpha_t \|\boldsymbol{x}\|_{V_t^{-1}}$ is the width of the confidence interval, indicating the uncertainty of $\langle \hat{\boldsymbol{\theta}}_t, \boldsymbol{x} \rangle$ (Zhang et al., 2016; Boyd & Vandenberghe, 2004).

## 2.2 MULTIOBJECTIVE BANDITS

The MOMAB problem was initially investigated by Drugan & Nowe (2013), who proposed two UCB-based algorithms that achieve regret bounds of $O(K \log T)$ under the Pareto regret metric and scalarized regret metric, respectively. The Pareto regret measures the cumulative distance between the obtained reward vectors and the Pareto optimal rewards, while the scalarized regret is the weighted regret of all objectives (Drugan & Nowe, 2013). To leverage environmental side information, Turgay et al. (2018) examined the multiobjective contextual bandit model, where the expected reward satisfies the Lipschitz condition with respect to contextual vectors. Lu et al. (2019) developed an algorithm with a Pareto regret bound of $\widetilde{O}(d\sqrt{T})$ for the multiobjective generalized linear bandit model. Another research direction focuses on designing algorithms from the perspective of best arm identification, with the primary goal of identifying Pareto optimal arms within a limited budget (Van Moffaert et al., 2014; Auer et al., 2016). Hüyük & Tekin (2021) is the only study for the multiobjective bandit problem under lexicographic ordering. They presented the PF-LEX algorithm for the MOMAB model, whose regret bound is $\widetilde{O}((KT)^{2/3})$ based on the priority-based regret metric (1). However, this result is inferior to existing single objective MAB algorithms, which attain a regret bound of $O(K \log T)$ (Lai & Robbins, 1985).

The intuitive idea to settle the lexicographically ordered issue for the multiobjective bandit model is to sequentially filter the arms according to the priority among objectives (Ehrgott, 2005; Hüyük & Tekin, 2021). To further illustrate this idea, we introduce the PF-LEX algorithm (Hüyük & Tekin, 2021). At each round $t$, PF-LEX first calculates confidence intervals for expected rewards through the historical trials. Specifically, the estimated reward of arm $a \in [K]$ in the $i$-th objective is given by $\hat{\mu}_t^i(a) = \sum_{\tau=1}^{t-1} y^i(a_\tau) \mathbb{I}(a_\tau = a)/N_t(a)$, where $a_\tau$ represents the arm played at round $\tau$ and $N_t(a)$ denotes the number of times arm $a$ has been played up to round $t$. Thus, the $i$-th confidence intervals for arm $a \in [K]$ is

$$\left[\hat{\mu}_t^i(a) - w_t(a), \hat{\mu}_t^i(a) + w_t(a)\right] \tag{10}$$

where $w_t(a) = \beta_t \sqrt{(1 + N_t(a))/N_t^2(a)}$ and $\beta_t = O(\sqrt{\log(Kmt)})$. Subsequently, PF-LEX either chooses the arm with a wide confidence interval to explore potentially better arms or selects the arm that is almost optimal in all objectives. Precisely, if some arm $a_t \in [K]$ satisfies $w_t(a_t) > \epsilon$ for a given criteria $\epsilon > 0$, PF-LEX chooses arm $a_t$. On the other hand, if $w_t(a) < \epsilon$ for all arms $a \in [K]$, PF-LEX filters the promising arms through the chain relation. Starting from $\mathcal{A}_t^0 = [K]$, PF-LEX operates as follows,

$$\hat{a}_t^i = \arg\max_{a \in \mathcal{A}_t^{i-1}} u_t^i(a), \mathcal{A}_t^i = \{a \in \mathcal{A}_t^{i-1} | a C_i \hat{a}_t^i\}, i \in [m]. \tag{11}$$

Here, $u_t^i(a) = \hat{\mu}_t^i(a) + w_t(a)$ and $a C_i \hat{a}_t^i$ denotes that arm $a$ and $\hat{a}_t^i$ are chained in the $i$-th objective, such that there exists a sequence of arms $\{a, b_1, b_2, \ldots, b_n, \hat{a}_t^i\} \subseteq [K]$, the $i$-th confidence intervals of adjacent arms are intersected. Finally, PF-LEX selects arm $\hat{a}_t^m$.

## 3 ALGORITHMS

In this section, we first extend the MOMAB algorithm proposed in Hüyük & Tekin (2021) to the MOSLB model as a warm-up and then provide an improved algorithm that achieves the almost optimal regret. Without loss of generality, we assume the arm vectors and inherent parameters are restricted in the unit sphere, such that $\|\boldsymbol{x}\| \le 1$ for any $\boldsymbol{x} \in \mathcal{D}_t, t \in [T]$ and $\|\boldsymbol{\theta}_*^i\| \le 1, i \in [m]$.

### 3.1 WARM-UP: STE$^2$LO

As a warm-up, we introduce the Single Trade-off between Exploration and Exploitation under Lexicographic Ordering (STE$^2$LO) algorithm, which is a simple extension of PF-LEX (Hüyük & Tekin, 2021). Given an input parameter $\epsilon > 0$, STE$^2$LO divides the decision-making operation at each round into two cases: pure exploration case and exploration-exploitation trade-off case.

We give a formal definition of the chain relation to facilitate our presentation. Given any arms $\boldsymbol{z}_1, \boldsymbol{z}_n \in \mathcal{D}_t$, we say that $\boldsymbol{z}_1$ and $\boldsymbol{z}_n$ are chained in the $i$-th objective, denoted by $\boldsymbol{z}_1 C_i \boldsymbol{z}_n$, if and only if there exists a sequence of arms $\{\boldsymbol{z}_1, \boldsymbol{z}_2, \ldots, \boldsymbol{z}_n\} \subset \mathcal{D}_t$ satisfying the condition that the

---

**Algorithm 1** Single Trade-off between Exploration and Exploitation under Lexicographic Ordering (STE$^2$LO)

---

**Input:** time horizon $T \in \mathbb{N}$, confidence parameter $\delta \in (0, 1)$, exploration criterion $\epsilon > 0$
1: Initialize $V_1 = I_d$ and $\hat{\boldsymbol{\theta}}_1^i = \mathbf{0}, i \in [m]$.
2: **for** $t = 1, 2, \ldots, T$ **do**
3:     Compute the estimated rewards and width of confidence intervals for any arm $\boldsymbol{x} \in \mathcal{D}_t$:
    $\hat{y}_t^i(\boldsymbol{x}) = \langle \hat{\boldsymbol{\theta}}_t^i, \boldsymbol{x} \rangle, \forall i \in [m], w_t(\boldsymbol{x}) = \gamma_t \|\boldsymbol{x}\|_{V_t^{-1}}$ where $\gamma_t = R\sqrt{d \ln(m(1+t)/\delta)} + 1$
4:     **if** $w_t(\boldsymbol{x}_t) > \epsilon$ for some $\boldsymbol{x}_t \in \mathcal{D}_t$ **then**
5:         Play the arm $\boldsymbol{x}_t$ and observe $[y_t^1, y_t^2, \ldots, y_t^m]$
6:     **else** $w_t(\boldsymbol{x}) \leq \epsilon \quad \forall \boldsymbol{x} \in \mathcal{D}_t$
7:         Initialize $\mathcal{D}_t^0 = \mathcal{D}_t$
8:         **for** $i = 1, 2, \ldots, m$ **do**
9:             $\hat{\boldsymbol{x}}_t^i = \arg\max_{\boldsymbol{x} \in \mathcal{D}_t^{i-1}} \hat{y}_t^i(\boldsymbol{x}) + w_t(\boldsymbol{x}), \mathcal{D}_t^i = \{\boldsymbol{x} \in \mathcal{D}_t^{i-1} | \boldsymbol{x} C_i \hat{\boldsymbol{x}}_t^i\}$
10:         **end for**
11:         Play the arm $\boldsymbol{x}_t = \hat{\boldsymbol{x}}_t^m$ and observe $[y_t^1, y_t^2, \ldots, y_t^m]$
12:     **end if**
13:     Update $V_{t+1} = V_t + \boldsymbol{x}_t \boldsymbol{x}_t^\top, X_{t+1} = [\boldsymbol{x}_\tau]_{\tau \in [t]}$ and $Y_{t+1}^i = [y_\tau^i]_{\tau \in [t]}, i \in [m]$
14:     Update the estimators $\hat{\boldsymbol{\theta}}_{t+1}^i = V_{t+1}^{-1} X_{t+1} Y_{t+1}^i, i \in [m]$
15: **end for**

---

confidence intervals of adjacent arms in the $i$-th objective are intersected, i.e., $[\ell_t^i(\boldsymbol{z}_j), u_t^i(\boldsymbol{z}_j)] \cap [\ell_t^i(\boldsymbol{z}_{j+1}), u_t^i(\boldsymbol{z}_{j+1})] \neq \emptyset, \forall j \in [n-1]$. Here, $\ell_t^i(\cdot)$ and $u_t^i(\cdot)$ denote the lower and upper confidence bounds for the $i$-th objective at epoch $t$, respectively.

At epoch $t$, considering that the agent receives $m$ values per epoch, SCE$^2$LO performs least square estimation on each value sequence to estimate the unknown parameters $\{\boldsymbol{\theta}_*^1, \boldsymbol{\theta}_*^2, \ldots, \boldsymbol{\theta}_*^m\}$, such that,

$$\hat{\boldsymbol{\theta}}_t^i = \arg\min_{\boldsymbol{\theta} \in \mathbb{R}^d} \|X_t \boldsymbol{\theta} - Y_t^i\|^2 + \|\boldsymbol{\theta}\|^2, i \in [m] \tag{12}$$

where $X_t = [\boldsymbol{x}_\tau]_{\tau \in [t-1]} \in \mathbb{R}^{(t-1) \times d}$ is the matrix of selected arms, and $Y_t^i = [y_\tau^i]_{\tau \in [t-1]} \in \mathbb{R}^{(t-1) \times 1}$ is the $i$-th historical rewards vector. Using a variant of the self-normalized bound for martingales (Abbasi-yadkori et al., 2011), the estimated rewards and the confidence interval width for any arm $\boldsymbol{x} \in \mathcal{D}_t$ can be calculated as

$$\hat{y}_t^i(\boldsymbol{x}) = \langle \hat{\boldsymbol{\theta}}_t^i, \boldsymbol{x} \rangle, w_t(\boldsymbol{x}) = \gamma_t \|\boldsymbol{x}\|_{V_t^{-1}}, i \in [m] \tag{13}$$

where $\gamma_t = R\sqrt{d \ln(m(1+t)/\delta)} + 1$ and $V_t = I_d + X_t X_t^\top$. Thus, for the arm $\boldsymbol{x} \in \mathcal{D}_t$, the confidence interval for the expected reward $\langle \boldsymbol{\theta}_*^i, \boldsymbol{x} \rangle$ is

$$[\ell_t^i(\boldsymbol{x}), u_t^i(\boldsymbol{x})] = [\hat{y}_t^i(\boldsymbol{x}) - w_t(\boldsymbol{x}), \hat{y}_t^i(\boldsymbol{x}) + w_t(\boldsymbol{x})]. \tag{14}$$

The wider confidence interval implies higher uncertainty in the estimate of expected reward, requiring the arm to be pulled to obtain more information. Therefore, if there exists some $\boldsymbol{x}_t \in \mathcal{D}_t$ that has a confidence interval wider than the input parameter $\epsilon$, i.e., $w_t(\boldsymbol{x}_t) > \epsilon$, SCE$^2$LO plays the arm $x_t$ as pure exploration. In contrast, if all arms have narrow confidence intervals, i.e., $w_t(\boldsymbol{x}) \leq \epsilon, \forall \boldsymbol{x} \in \mathcal{D}_t$, SCE$^2$LO tends to play the arm with the highest upper confidence bound in all objectives to balance exploration and exploitation. However, the arm with the highest upper confidence bound may vary for different objectives, preventing simultaneous maximization of all objectives. Considering the importance of different objectives, SCE$^2$LO filters the arms from the first objective to the last objective sequentially. More precisely, starts from $\mathcal{D}_t^0 = \mathcal{D}_t$, SCE$^2$LO filters the arm set through the filtering mechanism below,

$$\hat{\boldsymbol{x}}_t^i = \arg\max_{\boldsymbol{x} \in \mathcal{D}_t^{i-1}} \hat{y}_t^i(\boldsymbol{x}) + w_t(\boldsymbol{x}), \mathcal{D}_t^i = \{\boldsymbol{x} \in \mathcal{D}_t^{i-1} | \boldsymbol{x} C_i \hat{\boldsymbol{x}}_t^i\}, i \in [m] \tag{15}$$

where $\hat{\boldsymbol{x}}_t^i$ is the arm with the highest upper confidence bound in the $i$-th objective, and $\boldsymbol{x} C_i \hat{\boldsymbol{x}}_t^i$ selects the arms chained with the arm $\hat{\boldsymbol{x}}_t^i$. After filtering on the last objective, SCE$^2$LO plays the arm $\hat{\boldsymbol{x}}_t^m$

---

**Algorithm 2** Lexicographically Ordered Arm Filter (LOAF)

**Input:** arm set $\mathcal{D}_t$, scalarized paramter $\lambda$, maximum confidence intervals width $W$, upper confidence bound $u_t^i(\boldsymbol{x})$ for all $\boldsymbol{x} \in \mathcal{D}_t$ and $i \in [m]$.
1: Initialize the arm set $\mathcal{D}_t^0 = \mathcal{D}_t$
2: **for** $i = 1, 2, \ldots, m$ **do**
3:      $\hat{\boldsymbol{x}}_t^i = \arg\max_{\boldsymbol{x} \in \mathcal{D}_t^{i-1}} u_t^i(\boldsymbol{x})$
4:      $\mathcal{D}_t^i = \{\boldsymbol{x} \in \mathcal{D}_t^{i-1} | u_t^i(\boldsymbol{x}) \geq u_t^i(\hat{\boldsymbol{x}}^i) - (2 + 4\lambda + \ldots + 4\lambda^{i-1})W\}$
5: **end for**
6: Return the filtered arm set $\mathcal{D}_t^m$

---

and observes the reward vector $[y_t^1, y_t^2, \ldots, y_t^m]$. Finally, for $i \in [m]$, SCE$^2$LO updates the estimator from $\hat{\boldsymbol{\theta}}_t^i$ to $\hat{\boldsymbol{\theta}}_{t+1}^i$ with the updated contextual information matrix $X_{t+1}$ and historical rewards vector $Y_{t+1}^i$. The following theorem establishes the theoretical guarantees for the SCE$^2$LO algorithm.

**Theorem 1** *Suppose that (2) and (3) hold, and the arm sets are finite, i.e., $|\mathcal{D}_t| = K, \forall t \in [T]$. If STE$^2$LO is run with $\delta \in (0,1)$ and $\epsilon > 0$, then with probability at least $1 - \delta$, STE$^2$LO satisfies*

$$\widehat{R}^i(T) \leq 100\epsilon^{-2} d \ln T \left( R^2 d \ln \left( m(1+T)/\delta \right) + 1 \right) + 2KT\epsilon, \quad \forall i \in [m]$$

*where $\widehat{R}^i(T) = \sum_{t=1}^{T} \langle \boldsymbol{\theta}_*^i, \boldsymbol{x}_t^* - \boldsymbol{x}_t \rangle \mathbb{I}(\langle \boldsymbol{\theta}_*^j, \boldsymbol{x}_t^* \rangle = \langle \boldsymbol{\theta}_*^j, \boldsymbol{x}_t \rangle, 1 \leq j \leq i-1)$ is a priority-based regret.*

**Remark:** By setting the input parameter $\epsilon = d^{2/3}(KT)^{-1/3}$, Theorem 1 implies that STE$^2$LO can achieve a $\widetilde{O}((dKT)^{2/3})$ bound without requiring any prior knowledge. This bound matches the bound of the existing algorithm PF-LEX in terms of $K$ and $T$ (Hüyük & Tekin, 2021). STE$^2$LO allows the arm set $\mathcal{D}_t$ to vary during the learning process, a distinguishing feature from PF-LEX that lacks such flexibility. However, there are two limitations for this algorithm. First, STE$^2$LO is suboptimal as the lower regret bound for single objective SLB model is $\Omega(d\sqrt{T})$ (Dani et al., 2008). Second, the priority-based regret $\widehat{R}^i(T)$ relies on the indicator function $\mathbb{I}(\cdot)$, which only measures the performance of the first objective when $\langle \boldsymbol{\theta}_*^1, \boldsymbol{x}_t \rangle < \langle \boldsymbol{\theta}_*^1, \boldsymbol{x}_t^* \rangle$.

## 3.2 IMPROVED ALGORITHM: MTE$^2$LO

Although STE$^2$LO is straightforward, its regret bound is suboptimal even with the priority-based metric $\widehat{R}^i(T)$. In this section, we introduce an improved algorithm called MTE$^2$LO, which achieves the almost optimal bound on the general regret $R^i(T)$. To motivate the development of MTE$^2$LO, we first briefly explain why the simple algorithm STE$^2$LO is suboptimal.

One limitation of STE$^2$LO is the use of the chain relation $C_i$, which may result in the absence of the lexicographic optimal arm $\boldsymbol{x}_t^*$ from $\mathcal{D}_t^{i-1}$ to $\mathcal{D}_t^i$ for some $i \in [m]$. To illustrate this issue, we present a simple example with two objectives in Fig. 1. In this example, there are three arms, where the red point $\boldsymbol{x}_t^*$ represents the lexicographically optimal one. The square denotes the confidence intervals for the first and second objectives. Clearly, $\hat{\boldsymbol{x}}_t^1 = \hat{\boldsymbol{x}}_t^*$, and $\mathcal{D}_t^1$ contains both $\boldsymbol{x}_t^*$ and $\boldsymbol{x}$ since their confidence intervals for the first objective intersect. However, $\mathcal{D}_t^2$ loses $\boldsymbol{x}_t^*$ because $\hat{\boldsymbol{x}}_t^2 = \boldsymbol{x}$ and $\hat{\boldsymbol{x}}_t^2$ is not chained with $\boldsymbol{x}_t^*$ in the second objective. To remove this limitation, we observe that the confidence interval width for both objectives is equal for a fixed arm,

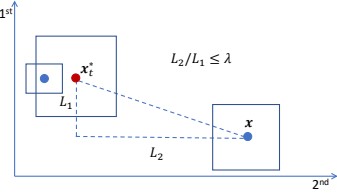

Figure 1: Motivation

and scaling the confidence interval of the second objective ensures the intersection of confidence intervals. Motivated by this observation, we design a novel Lexicographically Ordered Arm Filter (LOAF), which filters promising arms without losing the optimal arm, as detailed in Algorithm 2.

LOAF sequentially refines promising arms from the first objective to the last objective by the upper confidence bounds shown in the following equation,

$$u_t^i(\boldsymbol{x}) = \hat{y}_t^i(\boldsymbol{x}) + w_t(\boldsymbol{x}), i \in [m] \tag{16}$$

---

**Algorithm 3** Multiple Trade-off between Exploration and Exploitation under Lexicographic Ordering (MTE$^2$LO)

---

**Input:** time horizon $T \in \mathbb{N}$, scalarized parameter $\lambda$

1: Initialize $S = \lfloor \ln T \rfloor$, $V_0 = I_d$ and $\hat{\boldsymbol{\theta}}_0^i = \boldsymbol{0}$, $i \in [m]$.
2: **for** $t = 1, 2, \ldots, T$ **do**
3:     Compute the estimated rewards and width of confidence intervals for any arm $\boldsymbol{x} \in \mathcal{D}_t$:
    $\hat{y}_t^i(\boldsymbol{x}) = \langle \hat{\boldsymbol{\theta}}_t^i, \boldsymbol{x} \rangle, \forall i \in [m], w_t(\boldsymbol{x}) = \gamma_t \|\boldsymbol{x}\|_{V_t^{-1}}$ where $\gamma_t = R\sqrt{d \ln(m(1+t)/\delta)} + 1$
4:     Initialize $s = 1, \mathcal{D}_{t,1} = \mathcal{D}_t$
5:     **repeat**
6:         **if** $w_t(\boldsymbol{x}) \leq 1/\sqrt{T}$   $\forall \boldsymbol{x} \in \mathcal{D}_{t,s}$ **then**
7:             Invoke the Algorithm 2 to filter the promising arms $\mathcal{D}_{t,s}^m = \text{LOAF}\left(\lambda, 1/\sqrt{T}, \mathcal{D}_{t,s}\right)$
8:             Play the arm $\boldsymbol{x}_t = \arg\max_{\boldsymbol{x} \in \mathcal{D}_{t,s}^m} \hat{y}_t^m(\boldsymbol{x}) + w_t(\boldsymbol{x})$ and observe $[y_t^1, y_t^2, \ldots, y_t^m]$
9:         **else if** $w_t(\boldsymbol{x}_t) > 2^{-s}$ for some $\boldsymbol{x}_t \in \mathcal{D}_{t,s}$ **then**
10:            Play the arm $\boldsymbol{x}_t$ and observe $[y_t^1, y_t^2, \ldots, y_t^m]$
11:         **else** $w_t(\boldsymbol{x}) \leq 2^{-s}$   $\forall \boldsymbol{x} \in \mathcal{D}_{t,s}$
12:            Invoke the Algorithm 2 to filter the promising arms $\mathcal{D}_{t,s+1} = \text{LOAF}\left(\lambda, 2^{-s}, \mathcal{D}_{t,s}\right)$
13:            Update $s = s + 1$
14:         **end if**
15:     **until** an arm $\boldsymbol{x}_t$ is played.
16:     Update $V_{t+1} = V_t + \boldsymbol{x}_t \boldsymbol{x}_t^\top$, $X_{t+1} = [\boldsymbol{x}_\tau]_{\tau \in [t]}$ and $Y_{t+1}^i = [y_\tau^i]_{\tau \in [t]}, i \in [m]$
17:     Update the estimators $\hat{\boldsymbol{\theta}}_{t+1}^i = V_{t+1}^{-1} X_{t+1} Y_{t+1}^i, i \in [m]$
18: **end for**

---

where $\hat{y}_t^i(\boldsymbol{x})$ and $w_t(\boldsymbol{x})$ are the estimated reward and confidence interval width in (13). For the $i$-th objective, LOAF selects the most promising arms from the previous arm set $\mathcal{D}_t^{i-1}$ as follows,

$$\hat{\boldsymbol{x}}_t^i = \arg\max_{\boldsymbol{x} \in \mathcal{D}_t^{i-1}} u_t^i(\boldsymbol{x}) \tag{17}$$

where the initialized arm set $\mathcal{D}_t^0 = \mathcal{D}_t$. Then, LOAF retains the arms that are not far away from the arm $\hat{\boldsymbol{x}}_t^i$ in the $i$-th objective through the intersection of scalarized confidence intervals, i.e.,

$$\mathcal{D}_t^i = \left\{ \boldsymbol{x} \in \mathcal{D}_t^{i-1} | u_t^i(\boldsymbol{x}) \geq u_t^i(\hat{\boldsymbol{x}}_t^i) - (2 + 4\lambda + \ldots + 4\lambda^{i-1})W \right\} \tag{18}$$

where $\lambda$ is the scalarized parameter in the assumption (5), and $W$ is the maximum width of confidence intervals among the input arms. LOAF not only keeps the optimal arm in the returned arm set $\mathcal{D}_t^m$ but also ensures the expected rewards of arms in $\mathcal{D}_t^m$ are close to the optimal arm across all objectives. The following proposition supports this claim.

**Proposition 1** *For the algorithm LOAF, suppose that (5) holds and the expected rewards are contained within confidence intervals (14) with probability at least $1 - \delta$. If $\boldsymbol{x}_t^*$ is the optimal arm of the input arm set $\mathcal{D}_t$ and $W$ is the maximum width of confidence intervals for the input arms, then with probability at least $1 - \delta$, $\boldsymbol{x}_t^*$ belongs to the set $\mathcal{D}_t^m$, and*

$$\langle \boldsymbol{\theta}_*^i, \boldsymbol{x}_t^* - \boldsymbol{x} \rangle \leq 4(1 + \lambda + \ldots + \lambda^{i-1})W, i \in [m], \forall \boldsymbol{x} \in \mathcal{D}_t^m.$$

**Remark:** Proposition 1 demonstrates that LOAF returns an arm set $\mathcal{D}_t^m$ that contains the optimal arm $\boldsymbol{x}_t^*$. Meanwhile, it establishes a bound on the gap between the expected rewards of the optimal arm $\boldsymbol{x}_t^*$ and any other arm within the set $\mathcal{D}_t^m$. This bound is $O((1 + \lambda^{i-1})W)$, which increase exponentially as the index of objectives grows. However, most multiobjective problems typically involve two or three objectives (Deb & Jain, 2014; Li et al., 2015), thus $\lambda^{i-1}$ will not be extreme.

Another drawback of STE$^2$LO is its high trial consumption in the case $w_t(\boldsymbol{x}_t) > \epsilon$, which is a pure exploration case without any exploitation. To settle this issue, we divide the decision-making operation at each round into $S$ stages to make a delicate trade-off between exploration and exploitation. The proposed algorithm, Multiple Trade-off between Exploration and Exploitation under Lexicographic Ordering (MTE$^2$LO), is shown in Algorithm 3.

MTE$^2$LO adopts a framework similar to STE$^2$LO, but takes a more delicate decision-making process. At each time step $t$, MTE$^2$LO first calculates the estimated rewards and confidence interval

Table 1: Expected reward vectors for $\lambda = 0.1$ and $\lambda = 10$

| Arms | $\lambda = 0.1$ | $\lambda = 10$ |
|---|---|---|
| Arm 1 | $(0.42, -0.11, 0.06, -0.27, 0.41)$ | $(0.33, 0.50, 0.20, -0.23, -0.03)$ |
| Arm 2 | $(0.42, -0.24, -0.22, -0.48, 0.00)$ | $(0.33, 0.34, -0.18, 0.24, -0.13)$ |
| Arm 3 | $(0.17, -0.24, -0.40, -0.38, 0.34)$ | $(-0.06, 0.34, 0.20, -0.21, 0.31)$ |
| Arm 4 | $(-0.37, -0.07, -0.40, -0.27, -0.02)$ | $(0.28, 0.15, 0.20, -0.46, 0.18)$ |
| Arm 5 | $(-0.14, -0.12, -0.09, -0.27, 0.13)$ | $(0.24, 0.29, -0.18, -0.46, 0.03)$ |
| Arm 6 | $(0.22, -0.38, -0.26, -0.50, 0.13)$ | $(0.00, -0.29, -0.26, 0.43, 0.03)$ |
| Arm 7 | $(0.30, -0.18, -0.52, -0.75, 0.08)$ | $(-0.16, 0.45, 0.40, -0.22, 0.20)$ |
| Arm 8 | $(-0.06, -0.33, -0.56, -0.42, 0.10)$ | $(-0.22, -0.30, 0.03, -0.16, -0.15)$ |
| Arm 9 | $(-0.23, -0.30, -0.66, -0.33, 0.35)$ | $(0.19, -0.16, -0.18, -0.06, -0.14)$ |
| Arm 10 | $(0.40, -0.40, -0.14, -0.38, 0.07)$ | $(-0.35, -0.10, 0.40, 0.02, -0.08)$ |

width for each arm in $\mathcal{D}_t$, using the formula (13). Subsequently, MTE$^2$LO initiates a loop of $S$ stages to iteratively refine the promising arms, starting with $\mathcal{D}_{t,1} = \mathcal{D}_t$.

At each stage $s$, MTE$^2$LO first checks if the confidence interval widths for all arms in $\mathcal{D}_{t,s}$ is less than or equal to $1/\sqrt{T}$. If this is the case, MTE$^2$LO invokes the LOAF algorithm with the input arm set $\mathcal{D}_{t,s}$ and maximum confidence interval width $1/\sqrt{T}$, obtaining the promising arms set $\mathcal{D}_{t,s}^m$. Then, MTE$^2$LO plays the arm with the highest upper confidence bound at the $m$-th objective from $\mathcal{D}_{t,s}^m$ and records its rewards. Alternatively, if the confidence interval width of some arm in $\mathcal{D}_{t,s}$ exceeds $2^{-s}$, MTE$^2$LO plays this arm for exploration and records its rewards. Lastly, if the widths of all confidence intervals of the arms in $\mathcal{D}_{t,s}$ are less than or equal to $2^{-s}$, MTE$^2$LO applies the LOAF algorithm with the input arm set $\mathcal{D}_{t,s}$ and maximum confidence interval width $2^{-s}$ to update the promising arms set from $\mathcal{D}_{t,s}$ to $\mathcal{D}_{t,s+1}$. The last case balances exploration and exploitation because the maximum confidence interval width $2^{-s}$ promotes exploration, and the intersection of salarized confidence intervals in LOAF promises exploitation.

Let the total number of stages $S = \lfloor \ln T \rfloor$, then $2^{-S} < 1/\sqrt{T}$. Thus, MTE$^2$LO plays an arm before the decision-making loop ends. After playing an arm and observing its rewards, MTE$^2$LO updates the estimators $\hat{\boldsymbol{\theta}}_{t+1}^i, i \in [m]$. The following theorem guarantees the performance of MTE$^2$LO.

**Theorem 2** *Suppose that (2), (3) and (5) hold. If MTE$^2$LO is run with $\delta \in (0,1)$, then with probability at least $1 - \delta$, the regret of MTE$^2$LO satisfies*

$$R^i(T) \leq 8(1 + \lambda + \ldots + \lambda^{i-1}) \left( \sqrt{T} + 5d \ln T \left( R\sqrt{\ln\left(m(1+T)/\delta\right)} + 1 \right) \sqrt{T} \right), \forall i \in [m].$$

**Remark:** Theorem 2 states that MTE$^2$LO achieves the $\widetilde{O}((1 + \lambda^{i-1})d\sqrt{T})$ bound for the $i$-th objective, which is consistent with the optimal regret of single objective SLB algorithms in terms of the factors $d$ and $T$ (Dani et al., 2008; Abbasi-yadkori et al., 2011). In addition, the above theorem adopts the general regret $R^i(T)$, which measures the performance of each objective more accurately than the priority-based regret $\hat{R}^i(T)$. Hüyük & Tekin (2021) established an expected lower regret bound $\Omega(T^{2/3})$ for MOMAB under lexicographic ordering. This does not conflict with our result since we consider pseudo-regret instead of expected regret (Lattimore & Szepesvári, 2020).

## 4 EXPERIMENTS

In this section, we conduct experiments to present the empirical performance of our proposed algorithms. We adopt PF-LEX (Hüyük & Tekin, 2021) and OFUL (Abbasi-yadkori et al., 2011) as baselines, where PF-LEX is designed for MOMAB under lexicographic ordering, and OFUL is designed for single objective SLB model.

Following the existing experimental setup (Lu et al., 2019), we set the objective number $m = 5$ and feature dimension $d = 10$. The arm sets are fixed as $\mathcal{D}_t = \mathcal{D}$ for $t \geq 1$, and the arm number matches the feature dimension ($|\mathcal{D}| = 10$), which ensures that PF-LEX and our proposed algorithms

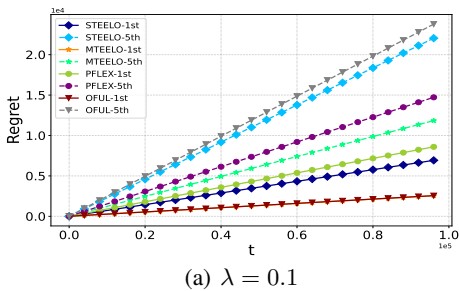 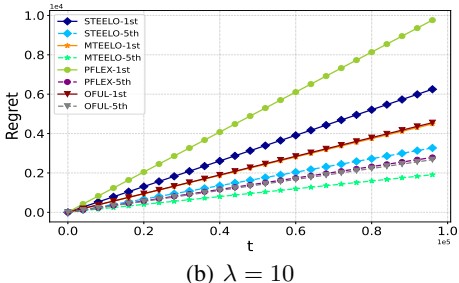

(a) $\lambda = 0.1$                    (b) $\lambda = 10$

Figure 2: Comparison of our algorithms versus OFUL and PF-LEX.

encounter the same number of unknown parameters. The coefficients $\boldsymbol{\theta}_*^i$ for $i \in [m]$ and arm $\boldsymbol{x} \in \mathcal{D}$ are uniformly sampled from the unit sphere[2]. Let the first sampled arm be the lexicographic optimal arm. We set the remaining nine arms to satisfy two conditions to distinguish the lexicographically ordered bandit problem from the single objective bandit problem. Firstly, there are two arms with equal expected rewards for each objective. Secondly, all arms satisfy the claim (5), and $\lambda$ is set as 0.1 and 10 respectively to demonstrate the performance of all algorithms across different problem difficulties. The expected reward vectors are summarized in Table 1. For the chosen arm $\boldsymbol{x}_t \in \mathcal{D}$, the reward for the $i$-th objective is given as $\langle \boldsymbol{\theta}_*^i, \boldsymbol{x}_t \rangle + \eta_t$, where $\eta_t$ is sampled from a Gaussian distribution with mean 0 and variance 1. We set $T = 10^5$ and $\delta = 0.01$ for all algorithms.

To reduce the randomness across the algorithms, we repeated each algorithm ten times and reported the average regret. The exploration parameter $\epsilon$ for STE$^2$LO and PF-LEX is set to $d^{2/3}(KT)^{-1/3}$ and $(KT)^{-1/3}$, respectively, which are theoretically optimal. In line with the common practice in bandit learning, we fine-tune the scaled parameters $\alpha_t, \beta_t$ and $\gamma_t$ of the confidence interval width in (9), (10), and (13), within the range of $[1e^{-3}, 1]$ (Jun et al., 2017; Lu et al., 2019).

Fig. 2 displays the general regret for the 1st and 5th objectives, where Fig. 2(a) and Fig. 2(b) present the results for the problem instances with $\lambda = 0.1$ and $\lambda = 10$, respectively. For $\lambda = 0.1$, OFUL performs the best in the first objective but performs worst in the fifth objective, as it is specifically designed for single objective bandit model. MTE$^2$LO exhibits comparable performance to OFUL in the first objective but significantly outperforms it in the fifth objective. STE$^2$LO performs slightly better than PF-LEX in the first objective but falls behind in the fifth objective. Both STE$^2$LO and PF-LEX are inferior to MTE$^2$LO. A interesting phenomenon is that all algorithms achieve better performance in the fifth objective than in the first when $\lambda = 10$, which is inconsistent with the regret bound in Theorem 2. This peculiarity can be attributed to the fact that most randomly sampled arms have higher expected rewards than the lexicographic optimal arm in the fifth objective for $\lambda = 10$.

## 5 CONCLUSION AND FUTURE WORK

We have investigated the MOSLB model under lexicographic ordering and presented two algorithms: STE$^2$LO and MTE$^2$LO. STE$^2$LO is straightforward and independent of prior knowledge, but its regret bound is suboptimal. The improved algorithm MTE$^2$LO achieves an almost optimal regret bound $\widetilde{O}((\lambda^{i-1} + 1)d\sqrt{T})$ for the $i$-th objective, $i \in [m]$. We extend the metric of lexicographically ordered multiobjective bandits from the priority-based regret (1) to the general regret (4), which more accurately evaluates the performance of algorithms. Our major novelties include a new arm filter and a multiple trade-off approach for exploration and exploitation. These techniques can be easily adapted to other bandit models, such as generalized linear bandits and Lipschitz bandits.

In the future, a challenging open problem is to develop an algorithm that is independent of the prior knowledge $\lambda$ and achieves the regret bound $\widetilde{O}(d\sqrt{T})$. Moreover, Theorem 2 demonstrates that objectives with lower priority have higher regret bounds, which contradicts the observed performance in Fig. 2(b). Thus, the regret bounds of low-priority objectives may be further reduced.

---

[2]The unit sphere is defined as the set $\{\boldsymbol{x} \in \mathbb{R}^d | \|\boldsymbol{x}\| \leq 1\}$.

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

## A    PROOF OF CLAIM (5)

The effectiveness of our proposed MTE$^2$LO algorithm depends on whether the claim (5) is true. In this section, we provide a detailed proof to show that the standard stochastic linear bandit model satisfies claim (5). As a result, our proposed algorithm, MTE$^2$LO, has the same scope of application as existing linear bandit algorithms (Auer, 2002; Abbasi-yadkori et al., 2011). To help with the understanding, we first prove the claim (5) for the first and second objectives.

**Proposition 2** *Suppose the arm sets $\{\mathcal{D}_1, \mathcal{D}_2, \ldots, \mathcal{D}_T\}$ are compact, then there exists some $\lambda_{(1,2)} \geq 0$, the expected rewards for the first and second objectives satisfy*

$$\langle \boldsymbol{\theta}_*^2, \boldsymbol{x} - \boldsymbol{x}_t^* \rangle \leq \lambda_{(1,2)} \cdot \langle \boldsymbol{\theta}_*^1, \boldsymbol{x}_t^* - \boldsymbol{x} \rangle \tag{19}$$

*for any $\boldsymbol{x} \in \mathcal{D}_t, t \in [T]$.*

**Proof.** In the case where $\lambda_{(1,2)}$ does not exist (i.e., $\lambda_{(1,2)} = +\infty$), there is at least one $t \in [T]$ that fails to satisfy inequality (19). For this special $t \in [T]$, let $S = \{\boldsymbol{x} \in \mathbb{R}^m | \langle \boldsymbol{\theta}_*^1, \boldsymbol{x} \rangle = \langle \boldsymbol{\theta}_*^1, \boldsymbol{x}_t^* \rangle\}$ represent the optimal line for the first objective, and let $\overline{\mathcal{D}}_t$ denote the closure of $\mathcal{D}_t$. The optimality of $\boldsymbol{x}_t^*$ guarantees that $\boldsymbol{x}_t^*$ lies at the endpoint of a line segment in $S \cap \mathcal{D}_t$. Since $\lambda_{(1,2)} = +\infty$, there must exist a sequence of points $\{\boldsymbol{z}_1, \boldsymbol{z}_2, \ldots, \boldsymbol{z}_n\} \subset \mathcal{D}_t$ such that $\lim_{n \to \infty} \langle \boldsymbol{\theta}_*^1, \boldsymbol{x}_t^* - \boldsymbol{z}_n \rangle = 0$ and $\lim_{n \to \infty} \langle \boldsymbol{\theta}_*^2, \boldsymbol{z}_n - \boldsymbol{x}_t^* \rangle = C > 0$. Let $\lim_{n \to \infty} \boldsymbol{z}_n = \boldsymbol{z}$, then $\langle \boldsymbol{\theta}_*^1, \boldsymbol{x}_t^* - \boldsymbol{z} \rangle = 0$ and $\langle \boldsymbol{\theta}_*^2, \boldsymbol{z} - \boldsymbol{x}_t^* \rangle = C > 0$. Therefore, $\boldsymbol{z}$ lies in $S \cap \overline{\mathcal{D}}_t$ and does not belong to $\mathcal{D}_t$. This implies that $\mathcal{D}_t \neq \overline{\mathcal{D}}_t$, which contradicts the standard setting that $\mathcal{D}_t$ is compact for the stochastic linear bandit model (Dani et al., 2008). As a result, $\lambda_{(1,2)}$ must exist in the standard stochastic linear bandit model. Hence, the proof of Proposition 2 is completed. $\square$

For the claim (5) with $m > 2$ objectives, the proof is similar to the Proposition 2 but with a different optimal line. Precisely, for a fixed $i \in [m]$, let the optimal line for previous $i - 1$ objectives is $S' = \{\boldsymbol{x} \in \mathbb{R}^m | \langle \boldsymbol{\theta}_*^j, \boldsymbol{x} \rangle = \langle \boldsymbol{\theta}_*^j, \boldsymbol{x}_t^* \rangle, j \in [i-1]\}$. If $\lambda = +\infty$, there must exist a sequence of points $\{\boldsymbol{z}_1', \boldsymbol{z}_2', \ldots, \boldsymbol{z}_n'\} \subset \mathcal{D}_t$ such that $\lim_{n \to \infty} \langle \boldsymbol{\theta}_*^1, \boldsymbol{x}_t^* - \boldsymbol{z}_n' \rangle = 0, \ldots, \lim_{n \to \infty} \langle \boldsymbol{\theta}_*^{i-1}, \boldsymbol{x}_t^* - \boldsymbol{z}_n' \rangle = 0$ and $\lim_{n \to \infty} \langle \boldsymbol{\theta}_*^i, \boldsymbol{z}_n' - \boldsymbol{x}_t^* \rangle = C > 0$. Let $\lim_{n \to \infty} \boldsymbol{z}_n' = \boldsymbol{z}'$, then $\langle \boldsymbol{\theta}_*^1, \boldsymbol{x}_t^* - \boldsymbol{z}' \rangle = 0, \ldots, \langle \boldsymbol{\theta}_*^{i-1}, \boldsymbol{x}_t^* - \boldsymbol{z}' \rangle = 0$ and $\langle \boldsymbol{\theta}_*^i, \boldsymbol{z}' - \boldsymbol{x}_t^* \rangle = C > 0$. Therefore, $\boldsymbol{z}'$ lies in $S' \cap \overline{\mathcal{D}}_t$ and does not belong to $\mathcal{D}_t$. This implies that $\mathcal{D}_t \neq \overline{\mathcal{D}}_t$, which contradicts the setting that $\mathcal{D}_t$ is compact. Thus, $\lambda$ must exist in standard stochastic linear bandit model. The proof is finished. $\square$

## B    PROOF OF THEOREM 1

To begin with the proof of Theorem 1, we present three lemmas that are crucial to our analysis. The first lemma guarantees the reliability of our constructed confidence intervals, the second lemma specifies the number of trials required for pure exploration in STE$^2$LO, and the third lemma gives the instantaneous regret for the exploration-exploitation trade-off case in STE$^2$LO.

**Lemma 1** *Suppose that (2) and (3) hold. If $\gamma_t = R\sqrt{d \ln(m(1+t)/\delta)} + 1$, then with probability at least $1 - \delta$, we have*

$$|\langle \hat{\boldsymbol{\theta}}_t^i, \boldsymbol{x} \rangle - \langle \boldsymbol{\theta}_*^i, \boldsymbol{x} \rangle| \leq \gamma_t \|\boldsymbol{x}\|_{V_t^{-1}}, \forall i \in [m], \forall t \geq 1$$

*for any $\boldsymbol{x} \in \mathcal{D}_t$.*

**Proof.** The sub-Gaussian property (3) allows us to expand the Confidence Ellipsoid theorem of Abbasi-yadkori et al. (2011) to a multiobjective context by using a union bound over the $m$ objectives, which states that for any $i \in [m]$ and $t \geq 1$, $\boldsymbol{\theta}_*^i$ lies in the confidence region

$$\mathcal{C}_t^i = \{\boldsymbol{\theta} \mid \|\boldsymbol{\theta} - \hat{\boldsymbol{\theta}}_t^i\|_{V_t} \leq \gamma_t\} \tag{20}$$

with probability at least $1 - \delta$, where $\gamma_t = R\sqrt{d \ln(m(1+t)/\delta)} + 1$.

Through the Lagrange method (Boyd & Vandenberghe, 2004), we calculate the upper confidence bound of $\langle \boldsymbol{\theta}_*^i, \boldsymbol{x} \rangle$ for a given arm $\boldsymbol{x} \in \mathcal{D}_t$ as

$$
\begin{aligned}
u_t^i(\boldsymbol{x}) &= \max_{\|\boldsymbol{\theta} - \hat{\boldsymbol{\theta}}_t^i\|_{V_t} \leq \gamma_t} \langle \boldsymbol{\theta}, \boldsymbol{x} \rangle \\
&= \max_{\|\boldsymbol{\theta}\|_{V_t} \leq \gamma_t} \langle \boldsymbol{\theta} + \hat{\boldsymbol{\theta}}_t^i, \boldsymbol{x} \rangle \\
&= \langle \hat{\boldsymbol{\theta}}_t^i, \boldsymbol{x} \rangle + \gamma_t \|\boldsymbol{x}\|_{V_t^{-1}}.
\end{aligned}
\tag{21}
$$

Similarly, the lower confidence bound of $\langle \boldsymbol{\theta}_*^i, \boldsymbol{x} \rangle$ for a given arm $\boldsymbol{x} \in \mathcal{D}_t$ is given by

$$
\ell_t^i(\boldsymbol{x}) = \langle \hat{\boldsymbol{\theta}}_t^i, \boldsymbol{x} \rangle - \gamma_t \|\boldsymbol{x}\|_{V_t^{-1}}.
\tag{22}
$$

Thus, we can conclude that with probability at least $1 - \delta$, for any $\boldsymbol{x} \in \mathcal{D}_t$, the following inequality holds for all $i \in [m]$ and $t \geq 1$,

$$
|\langle \hat{\boldsymbol{\theta}}_t^i, \boldsymbol{x} \rangle - \langle \boldsymbol{\theta}_*^i, \boldsymbol{x} \rangle| \leq \gamma_t \|\boldsymbol{x}\|_{V_t^{-1}}.
\tag{23}
$$

The proof of Lemma 1 is finished. $\qquad\square$

**Lemma 2** *In STE$^2$LO, suppose $\psi(t, \epsilon) = \{\tau \in [t] | w_\tau(\boldsymbol{x}_\tau) > \epsilon\}$ with $w_\tau(\boldsymbol{x}_\tau) = \gamma_\tau \|\boldsymbol{x}_\tau\|_{V_\tau^{-1}}$, then*

$$
|\psi(T, \epsilon)| \leq 50\epsilon^{-2} d \ln T \left(R^2 d \ln(m(1+T)/\delta) + 1\right).
$$

**Proof.** Firstly, we use Lemma 3 from Chu et al. (2011) to obtain

$$
\sum_{t \in \psi(T, \epsilon)} \|\boldsymbol{x}_t\|_{\widetilde{V}_t^{-1}} \leq 5\sqrt{d |\psi(T, \epsilon)| \ln |\psi(T, \epsilon)|}
\tag{24}
$$

where $\widetilde{V}_t = I_d + \sum_{\tau \in \psi(t-1, \epsilon)} \boldsymbol{x}_\tau \boldsymbol{x}_\tau^\top$. Since $V_t = I_d + \sum_{\tau \in [t-1]} \boldsymbol{x}_\tau \boldsymbol{x}_\tau^\top$, we can observe that $\|\boldsymbol{x}_t\|_{V_t^{-1}} \leq \|\boldsymbol{x}_t\|_{\widetilde{V}_t^{-1}}$. Therefore, we get

$$
\sum_{t \in \psi(T, \epsilon)} \|\boldsymbol{x}_t\|_{V_t^{-1}} \leq 5\sqrt{d |\psi(T, \epsilon)| \ln |\psi(T, \epsilon)|}.
\tag{25}
$$

Use the fact that $\gamma_t \|\boldsymbol{x}_t\|_{V_t^{-1}} > \epsilon$ for $t \in \psi(T, \epsilon)$, we can derive

$$
\epsilon |\psi(T, \epsilon)| \leq 5\gamma_T \sqrt{d |\psi(T, \epsilon)| \ln |\psi(T, \epsilon)|}.
\tag{26}
$$

Simplify the above inequality and take $\gamma_T = R\sqrt{d \ln(m(1+T)/\delta)} + 1$ into it, we obtain

$$
|\psi(T, \epsilon)| \leq 25\epsilon^{-2} \gamma_T^2 d \ln T \leq 50\epsilon^{-2} d \ln T (R^2 d \ln(m(1+T)/\delta) + 1).
\tag{27}
$$

This completes the proof. $\qquad\square$

**Lemma 3** *In STE$^2$LO, suppose $\widetilde{\psi}(t, \epsilon) = \{\tau \in [t] | w_\tau(\boldsymbol{x}_\tau) \leq \epsilon\}$ with $w_\tau(\boldsymbol{x}_\tau) = \gamma_\tau \|\boldsymbol{x}_\tau\|_{V_\tau^{-1}}$, then with probability at least $1 - \delta$, for any $t \in \widetilde{\psi}(T, \epsilon)$*

$$
\langle \boldsymbol{\theta}_*^i, \boldsymbol{x}_t^* - \boldsymbol{x}_t \rangle \cdot \mathbb{I}\left(\langle \boldsymbol{\theta}_*^j, \boldsymbol{x}_t^* \rangle = \langle \boldsymbol{\theta}_*^j, \boldsymbol{x}_t \rangle, j \in [i-1]\right) \leq 2K\epsilon, i \in [m].
$$

**Proof.** In the first case where $\mathbb{I}\left(\langle \boldsymbol{\theta}_*^j, \boldsymbol{x}_t^* \rangle = \langle \boldsymbol{\theta}_*^j, \boldsymbol{x}_t \rangle, j \in [i-1]\right) = 0$, Lemma 3 holds obviously.

Then, we analyze the second case $\langle \boldsymbol{\theta}_*^j, \boldsymbol{x}_t^* \rangle = \langle \boldsymbol{\theta}_*^j, \boldsymbol{x}_t \rangle, j \in [i-1]$. According to the definition of upper confidencce bound and lower confidence bound, we can easily get

$$
\langle \boldsymbol{\theta}_*^i, \boldsymbol{x}_t^* \rangle - \langle \boldsymbol{\theta}_*^i, \boldsymbol{x}_t \rangle \leq u_t^i(\boldsymbol{x}_t^*) - \ell_t^i(\boldsymbol{x}_t)
\tag{28}
$$

holds with probability at least $1 - \delta$.

Use the induced method, we can prove that $\boldsymbol{x}_t^* \in \mathcal{D}_t^{i-1}$ in the second case. Precisely, for $i = 1$, it is obvious that $\boldsymbol{x}_t^* \in \mathcal{D}_t^0$ since $\mathcal{D}_t^0 = \mathcal{D}_t$. Then, we assume that $\boldsymbol{x}_t^* \in \mathcal{D}_t^{i-2}$. Next, we focus on proving $\boldsymbol{x}_t^* \in \mathcal{D}_t^{i-1}$. Since $\langle \boldsymbol{\theta}_*^{i-1}, \boldsymbol{x}_t^* \rangle = \langle \boldsymbol{\theta}_*^{i-1}, \boldsymbol{x}_t \rangle$, $\boldsymbol{x}_t^*$ and $\boldsymbol{x}_t$ are chained in the $(i-1)$-th objective. The

filter method $\mathcal{D}_t^{i-1} = \{x \in \mathcal{D}_t^{i-2} | x C_{i-1} \hat{x}_t^{i-1}\}$ shows that $x_t$ is chained with $\hat{x}_t^{i-1}$. Since the chain relation satisfies the transitive property, $x_*^i$ is chained with $\hat{x}_t^{i-1}$, thus $x_t^* \in \mathcal{D}_t^{i-1}$.

According to $\hat{x}_t^i = \arg\max_{x \in \mathcal{D}_t^{i-1}} u_t^i(x)$, we get that

$$u_t^i(x_t^*) - \ell_t^i(x_t) \leq u_t^i(\hat{x}_t^i) - \ell_t^i(x_t). \tag{29}$$

Combining equations (28) and (29) yeilds

$$\langle \theta_*^i, x_t^* \rangle - \langle \theta_*^i, x_t \rangle \leq u_t^i(\hat{x}_t^i) - \ell_t^i(x_t). \tag{30}$$

In the decision rounds $\widetilde{\psi}(T, \epsilon)$, it can be observed that all arms have confidence intervals that are smaller than $\epsilon$, and given that there are $K$ arms in total, we can conclude that $u_t^i(\hat{x}_t^i) - \ell_t^i(x_t) \leq 2K\epsilon$. This inequality can be substituted into equation (30) to complete the proof. $\square$

Take the Lemma 2 and Lemma 3 into $\widehat{R}^i(T)$, we get that with probability at least $1 - \delta$,

$$
\begin{aligned}
\widehat{R}^i(T) &= \sum_{t=1}^T \langle \theta_*^i, x_t^* - x_t \rangle \mathbb{I}(\langle \theta_*^j, x_t^* \rangle = \langle \theta_*^j, x_t \rangle, 1 \leq j \leq i-1) \\
&= \sum_{t \in \psi(T, \epsilon)} \langle \theta_*^i, x_t^* - x_t \rangle \mathbb{I}(\langle \theta_*^j, x_t^* \rangle = \langle \theta_*^j, x_t \rangle, 1 \leq j \leq i-1) \\
&\quad + \sum_{t \in \widetilde{\psi}(T, \epsilon)} \langle \theta_*^i, x_t^* - x_t \rangle \mathbb{I}(\langle \theta_*^j, x_t^* \rangle = \langle \theta_*^j, x_t \rangle, 1 \leq j \leq i-1) \\
&\leq 2|\psi(T, \epsilon)| + 2KT\epsilon \\
&\leq 100\epsilon^{-2} d \ln T \left(R^2 d \ln(m(1+T)/\delta) + 1\right) + 2KT\epsilon.
\end{aligned}
\tag{31}
$$

The proof of Theorem 1 is completed. $\square$

## C  PROOF OF PROPOSITION 1

To prove the Proposition 1, we use induction method. For $i = 1$, we note that

$$u_t^1(x_t^*) \geq \langle \theta_*^1, x_t^* \rangle \geq \langle \theta_*^1, \hat{x}_t^1 \rangle \geq u_t^1(\hat{x}^1) - 2W, \tag{32}$$

which confirms that $x_t^* \in \mathcal{D}_t^1$ since $\mathcal{D}_t^1 = \{x \in \mathcal{D}_t^0 | u_t^1(x) \geq u_t^1(\hat{x}_t^1) - 2W\}$ and $x_t^* \in \mathcal{D}_t^0$. Then, based on the filtered method in $\mathcal{D}_t^1$, we get that for any $x \in \mathcal{D}_t^1$,

$$\langle \theta_*^1, x \rangle \geq u_t^1(x) - 2W \geq u_t^1(\hat{x}_t^1) - 4W \geq u^1(x_t^*) - 4W \geq \langle \theta_*^1, x_t^* \rangle - 4W, \tag{33}$$

which indicates $\langle \theta_*^1, x_t^* - x \rangle \leq 4W$.

Next, we prove that if $x_t^* \in \mathcal{D}_t^{i-1}$ then $x_t^* \in \mathcal{D}_t^i$, and if $\langle \theta_*^j, x_t^* - x \rangle \leq 4(1 + \lambda + \ldots + \lambda^{j-1})W$ for $j \in [i-1]$ then $\langle \theta_*^i, x_t^* - x \rangle \leq 4(1 + \lambda + \ldots + \lambda^{i-1})W$ for any $x \in \mathcal{D}_t^i$.

According to inequality (5), it is evident that $\langle \theta_*^i, x_t^* \rangle \geq \langle \theta_*^i, \hat{x}_t^i \rangle - \lambda \cdot \max_{j \in [i-1]} \langle \theta_*^j, x_t^* - \hat{x}_t^i \rangle$. Moreover, considering that the maxmium confidence interval width is $W$, $u_t^i(x_t^*) \geq \langle \theta_*^i, x_t^* \rangle$ and $\langle \theta_*^i, \hat{x}_t^i \rangle \geq u_t^i(\hat{x}_t^i) - 2W$ hold. Thus, we get that

$$u_t^i(x_t^*) \geq u_t^i(\hat{x}_t^i) - 2W - (4\lambda + 4\lambda^2 + \ldots + 4\lambda^{i-1})W, \tag{34}$$

which indicates $x_t^* \in \mathcal{D}_t^i$ since $\mathcal{D}_t^i = \{x \in \mathcal{D}_t^{i-1} | u_t^i(x) \geq u_t^i(\hat{x}^i) - (2 + 4\lambda + \ldots + 4\lambda^{i-1})W\}$. With $x_t^* \in \mathcal{D}_t^i$ and the filtered method in $\mathcal{D}_t^i$, we can derive that for any $x \in \mathcal{D}_t^i$,

$$
\begin{aligned}
\langle \theta_*^i, x \rangle &\geq u_t^i(x) - 2W \\
&\geq u_t^i(\hat{x}_t^i) - 4W - (4\lambda + 4\lambda^2 + \ldots + \lambda^{i-1})W \\
&\geq u_t^i(x_t^*) - 4(1 + \lambda + \ldots + \lambda^{i-1})W \\
&\geq \langle \theta_*^i, x_t^* \rangle - 4(1 + \lambda + \ldots + \lambda^{i-1})W.
\end{aligned}
\tag{35}
$$

Thus we get $\langle \theta_*^i, x_t^* - x \rangle \leq 4(1 + \lambda + \ldots + \lambda^{i-1})W$. The proof is finished. $\square$

# D  PROOF OF THEOREM 2

One of the most important advantage of MTE$^2$LO is the multiple trade-off between exploration and exploitation through the proceeding of multiple stages in the decision-making process. We provide the following lemma to specify the number of trials at each stage.

**Lemma 4** *In MTE$^2$LO, suppose $\psi_s(t) = \{\tau \in [t] | \boldsymbol{x}_\tau$ is played in the **else if** case: $w_\tau(\boldsymbol{x}_\tau) > 2^{-s}\}$, where $w_\tau(\boldsymbol{x}_\tau) = \gamma_\tau \|\boldsymbol{x}_\tau\|_{V_\tau^{-1}}$, then for all $s \in [S]$,*

$$|\psi_s(T)| \leq 5 \cdot 2^s \left( R\sqrt{d\ln(m(1+T)/\delta)} + 1 \right) \sqrt{d|\psi_s(T)|\ln|\psi_s(T)|}.$$

**Proof.** Firstly, we use Lemma 3 from Chu et al. (2011) to obtain

$$\sum_{t \in \psi_s(T)} \|\boldsymbol{x}_t\|_{\widetilde{V}_t^{-1}} \leq 5\sqrt{d|\psi_s(T)|\ln|\psi_s(T)|} \tag{36}$$

where $\widetilde{V}_t = I_d + \sum_{\tau \in \psi_s(t-1)} \boldsymbol{x}_\tau \boldsymbol{x}_\tau^\top$. Since $V_t = I_d + \sum_{\tau \in [t-1]} \boldsymbol{x}_\tau \boldsymbol{x}_\tau^\top$, we can observe that $\|\boldsymbol{x}_t\|_{V_t^{-1}} \leq \|\boldsymbol{x}_t\|_{\widetilde{V}_t^{-1}}$. Therefore, we get

$$\sum_{t \in \psi_s(T)} \|\boldsymbol{x}_t\|_{V_t^{-1}} \leq 5\sqrt{d|\psi_s(T)|\ln|\psi_s(T)|}. \tag{37}$$

Using the fact that $\gamma_t \|\boldsymbol{x}_t\|_{V_t^{-1}} > 2^{-s}$ for $t \in \psi_s(T)$, we can derive

$$2^{-s}|\psi_s(T)| \leq 5\gamma_T \sqrt{d|\psi_s(T)|\ln|\psi_s(T)|}. \tag{38}$$

Taking $\gamma_T = R\sqrt{d\ln(m(1+T)/\delta)} + 1$ into above equation tells that

$$|\psi_s(T)| \leq 5 \cdot 2^s \left( R\sqrt{d\ln(m(1+T)/\delta)} + 1 \right) \sqrt{d|\psi_s(T)|\ln|\psi_s(T)|}. \tag{39}$$

The proof of Lemma 4 is finished. $\qquad\square$

Let $\psi_0(T) = [T] \setminus \bigcup_{s \in [S]} \psi_s(T)$ denote the trials whose confidence interval width is less than or equal to $1/\sqrt{T}$. Then, the regret for the $i$-the objective can be rewritten as

$$R^i(T) = \sum_{t \in \psi_0(T)} \langle \boldsymbol{\theta}_*^i, \boldsymbol{x}_t^* - \boldsymbol{x}_t \rangle + \sum_{s=1}^{S} \sum_{t \in \psi_s(T)} \langle \boldsymbol{\theta}_*^i, \boldsymbol{x}_t^* - \boldsymbol{x}_t \rangle \tag{40}$$

The trials in $\psi_0(T)$ are filtered by LOAF with maximum width $W = 1/\sqrt{T}$ and the trials in $\psi_s(T)$ are filtered by LOAF with maximum width $W = 2 \cdot 2^{-s}$. Based on Proposition 1, we have that

$$\sum_{t \in \psi_0(T)} \langle \boldsymbol{\theta}_*^i, \boldsymbol{x}_t^* - \boldsymbol{x}_t \rangle \leq 4(1 + \lambda + \ldots + \lambda^{i-1})|\psi_0(T)|/\sqrt{T} \tag{41}$$

and

$$\sum_{t \in \psi_s(T)} \langle \boldsymbol{\theta}_*^i, \boldsymbol{x}_t^* - \boldsymbol{x}_t \rangle \leq 4(1 + \lambda + \ldots + \lambda^{i-1})2 \cdot 2^{-s}|\psi_s(T)|. \tag{42}$$

Thus, the regret for the $i$-the objective can be bounded by

$$R^i(T) \leq 4(1 + \lambda + \ldots + \lambda^{i-1}) \left( |\psi_0(T)|/\sqrt{T} + \sum_{s=1}^{S} 2 \cdot 2^{-s}|\psi_s(T)| \right) \tag{43}$$

By Lemma 4, we obtain

$$\begin{aligned}
\sum_{s=1}^{S} 2 \cdot 2^{-s}|\psi_s(T)| &\leq 10 \left( R\sqrt{d\ln(m(1+T)/\delta)} + 1 \right) \sum_{s=1}^{S} \sqrt{d|\psi_s(T)|\ln|\psi_s(T)|} \\
&\leq 10d \left( R\sqrt{\ln(m(1+T)/\delta)} + 1 \right) \sqrt{ST\ln T} \\
&\leq 10d\ln T \left( R\sqrt{\ln(m(1+T)/\delta)} + 1 \right) \sqrt{T}.
\end{aligned} \tag{44}$$

Taking equation (44) into equation (43) shows that

$$R^i(T) \leq 8(1 + \lambda + \ldots + \lambda^{i-1}) \left( \sqrt{T} + 5d\ln T \left( R\sqrt{\ln(m(1+T)/\delta)} + 1 \right) \sqrt{T} \right). \tag{45}$$

The proof is finished. $\qquad\square$

