# OpenReview forum: "Multiobjective Stochastic Linear Bandits under Lexicographic Ordering"
_ICLR.cc/2024/Conference — Submitted to ICLR 2024_

### Official Review · Reviewer_vLB6 · 2023-10-28

**Soundness:** 3 good
**Presentation:** 3 good
**Contribution:** 2 fair
**Rating:** 5
**Confidence:** 3

**Summary:**

This paper studied the linear setting of multiobjective stochastic bandits under lexicographic ordering. It proposed two algorithms: STE$^2$LO and MTE$^2$LO; the latter one is better. The paper provided a detailed description of the setting and explained how they proposed the two algorithms. The proposed algorithms and those existing algorithms are compared both theoretically and numerically.

================

After rebuttal: Thanks for the response. I just increased the score to 5.

**Strengths:**

1. The paper first introduced naive linear setting and then the multiobjective one under lexicographic ordering. Moreover, it introduced the STE$^2$LO algorithm before the MTE$^2$LO algorithm, which helps readers to understand the better but more complicated MTE$^2$LO algorithm.
1. The paper provides a detailed formulation of the multiobjective linear stochastic bandits under lexicographic ordering.
1. Numerical experiments are conducted to evaluate the performance of proposed algorithms.

**Weaknesses:**

Notation:
1. There seems to be a major typo among the whole writeup: do both $m$ and $d$ stand for the dimension of vectors? If so, the author(s) should consider to unify the notation. If not, may you clarify what do they mean individually?

Contribution of the work:
1. As no lower bound is provided, is there still space to improve the algorithm? What is the difficulty to provide a lower bound? As Huyuk & Tekin (2021) provided a lower bound for the priority-based regret, is it possible to derive a lower bound on the general regret similarly? Why are we interested in the general regret?
1. The STE$^2$LO and MTE$^2$LO algorithms seem to have similar structures to the SupLinRel and SupLinUCB algorithms.
    1. The two proposed algorithms seem to be adopted versions to the multi-objective setting. May the author(s) clarify what are the analytical challenge?
    1. For this reason, the author(s) may consider to condense the description of the two algorithms.
1. The fundamental parameter $\lambda$ is assumed to be known in this work. I appreciate the elaboration of importance of $\lambda$ at the bottom of page 2. However, is it reasonable to assume that $\lambda$ is known?
    1. The abstract states that 'This model has various real-world scenarios, including water resource planning and radiation treatment for cancer patients.' However, a more detailed description of real-life application is appreciated. I surmise that this may help me to understand why we can assume that $\lambda$ is known.
     1. In the example given in the first paragraph, what does 'click-conversion rate' mean?
1. Moreover, experiments with real-world data may make the efficiency of proposed algorithms more convincing.

Presentation:
1. I appreciate that the author(s) discussed many existing results in multiobjective bandits and related settings. However, as different settings/definitions of regret are considered, I think a table would provide a much clearer comparison.
1. I do appreciate that the author(s) introduced the standard linear setting first and the multiobjective one after that. However, the notations are defined throughout the first 3 pages. I think that a notation table may help readers to find notations.

**Questions:**

Except for the questions in the 'Weaknesses' section, here are more suggestions:
1. In the second paragraph of the introduction, it states that 'Therefore, if the evaluation criterion is Pareto regret, the agent can select any of the m objectives to optimize and ignore other objectives, which is unreasonable.'
This statement is indeed strong. The author(s) may consider to explain this point a little bit more.
1. Minor suggestion in the second paragraph of  Section 3.1: 'We provide a formal definition of the chain relation to facilitate our presentation' may be a better expression than 'We give a formal definition of the chain relation to facilitate our presentation'.

---

> ### Author Response · Authors · 2023-11-17
> **Response to Reviewer vLB6 (Part I)**
>
> Many thanks for your comments and suggestions. We have carefully considered your concerns and our responses are provided as follows.
>
> ----
>
> **Q1: There seems to be a major typo among the whole writeup: do both $m$ and $d$ stand for the dimension of vectors? If so, the author(s) should consider to unify the notation. If not, may you clarify what do they mean individually?**
>
> We investigate the multiobjective stochastic linear bandit problem. $m$ denotes the number of objectives, and $d$ is the dimension of arm vector. We will revise our paper to make the notation clear in the final version.
>
> ----
>
> **Q2: As no lower bound is provided, is there still space to improve the algorithm?**
>
> We are working hard to construct a lower bound for the problem investigated in our paper, thus we are unable to provide a definitive answer for this question now.
>
> ----
>
> **Q3: What is the difficulty to provide a lower bound? As Huyuk $\\&$ Tekin (2021) provided a lower bound for the priority-based regret, is it possible to derive a lower bound on the general regret similarly?**
>
> The lower bound for single objective linear bandits is established as $\Omega(d\sqrt{T})$ by Dani et al. (2008). The lower bound for the problem of our paper is supposed to be $\Omega(\lambda^{i-1}d\sqrt{T})$ for the $i$-th objective. The main challenge lies in the parameter $\lambda$. To construct a lower bound for any algorithm, we need to design an example that demonstrates a regret bound larger than $\Omega(\lambda^{i-1}d\sqrt{T})$ for the $i$-th objective. This task is both creative and demanding. The lower bound proposed by Huyuk $\&$ Tekin (2021) is $\Omega(T^{2/3})$, which solely depends on $T$ and does not involve $d$ or $\lambda$. Thus, the approach to constructing the lower bound $\Omega(\lambda^{i-1}d\sqrt{T})$ differs significantly from that of Huyuk $\&$ Tekin (2021).
>
> ----
>
> **Q4: Why are we interested in the general regret?**
>
> The priority-based regret $\widehat{R}^i(T)$ of Hüyük  $\\&$  Tekin (2021) is defined as follows:
> $$
> \widehat{R}^i(T)=\sum_{t=1}^T\langle \theta_\*^i, x_t^\* - x_t\rangle\mathbb{I}(\langle \theta_\*^j, x_t^\*\rangle = \langle \theta_\*^j, x_t\rangle, 1\leq j\leq i-1), i=1,2,\dots,m
> $$
> where $x_t^\*$ denotes the optimal arm in $\mathcal{D}\_t$ according to the lexicographic order. Note that the priority-based regret $\widehat{R}^i(T)$ relies on the indicator function $\mathbb{I}(\cdot)$, which cannot accumulate the instantaneous gap $\langle \theta_\*^i, x_t^\* - x_t\rangle$ for $i\geq 2$ if $\langle \theta_\*^1, x_t\rangle < \langle \theta_\*^1, x_t^\*\rangle$. Thus, $\widehat{R}^i(T)$ can not accurately measure the performance of agent for $i\geq2$.
>
> Our proposed regret $R^i(T)$ is a commonly used metric in the single-objective bandit field (Auer, 2002; Bubeck $\\&$ Cesa-Bianchi, 2012; Lattimore  $\\&$ Szepesvári, 2020), and we simply extend it to $m$ objectives, such that
> $$
> R^i(T)=\sum_{t=1}^T\langle \theta_\*^i, x_t^\* - x_t\rangle, i=1,2,\dots,m.
> $$
> $R^i(T)$ removes the indicator function, enabling the accumulation of the rewards gap between the selected arm $x_t$ and the lexicographically optimal arm $x_t^\*$ independently for all objectives $i=1,2,\ldots,m$. This makes it a more reliable metric for reflecting the algorithms' performance.

---

> ### Author Response · Authors · 2023-11-17
> **Response to Reviewer vLB6 (Part II)**
>
> **Q5: The STE$^2$LO and MTE$^2$LO algorithms seem to have similar structures to the SupLinRel and SupLinUCB algorithms. The two proposed algorithms seem to be adopted versions to the multi-objective setting. May the author(s) clarify what are the analytical challenge?**
>
> Although our algorithm MTE$^2$LO has a similar structure to SupLinRel and SupLinUCB, the purpose of applying this framework is completely different. SupLinRel and SupLinUCB utilize this sophisticated framework to guarantee independence among rewards, while we use this framework to ensure that the confidence intervals of the arm input to Algorithm 2 (LOAF) are bounded. The effectiveness of LOAF requires the confidence interval of the input arms to be bounded as we stated in Proposition 1.
>
> Our analysis faces two challenges. The first challenge lies in proving Proposition 1, where we not only demonstrate that the novel arm filter LOAF can maintain the lexicographic optimal arm, but also provide a bound to measure the quality of the filtered arms. Mathematical induction is an important tool to prove Proposition~1. The second challenge is analyzing the multi-stage decision-making process (Step 4 to Step 15) in Algorithm 3, which delicately balance the exploration and exploitation. Here, we give a proof sketch for the analysis of multi-stage decision-making process and more details can be found in Appendix D.
>
> Let $\psi_s(T)$ be the set composed of the number of arms with confidence intervals larger than $2^{-s}$. The first step is to prove that $|\psi_s(T)|$ is bounded by $\tilde{O}(2^s\sqrt{\psi_s(T)})$, as shown in Lemma 4. Note that the arm selected in the $s$-th stage is filtered by LOAF in the $(s-1)$-th stage, thus Proposition~1 shows that the instantaneous regret of these arms is bounded by $\tilde{O}(2^{-(s-1)})$. The last step is to accumulate all instantaneous regret, $\sum_{s=1}^{S}2^{-(s-1)}\cdot2^{s}\sqrt{\psi_s(T)}=\tilde{O}\sqrt{T}$. Further discussion on these challenges will be provided in the final version.
>
> ----
>
> **Q6: The fundamental parameter $\lambda$ is assumed to be known in this work. I appreciate the elaboration of importance of $\lambda$ at the bottom of page 2. However, is it reasonable to assume that $\lambda$ is known?**
>
> The dependence on $\lambda$ is indeed a limitation, as mentioned in the conclusion and future work section. However, in most multi-objective scenarios, it is possible to estimate $\lambda$. As we have discussed under Equation (5), $\lambda$ can be estimated by measuring how different objectives change as the decision varies. To further illustrate this, we provide a simple example with two objectives.
>
> The expected reward functions are denoted as $\mu^1(x)$ and $\mu^2(x)$, which is defined as $\mu^1(x)=1-\min_{p\in\\{0.4,0.8\\}}|x-p|$ and $\mu^2(x)=1-2|x-0.3|$ (Here we do not take a linear function so as to highlight the change rates of  expected reward functions). The optimal arms for the first objective are $\\{0.4, 0.8\\}$, and the lexicographic optimal arm is $0.4$. In this example, the value of $\lambda$ is 2, which is the ratio of the change rates of $\mu^1(x)$ and $\mu^2(x)$.
>
> Another point worth mentioning is that it is not necessary to know the **precise** ratio of the change rates. Any value **greater** than the ratio of the change rates is sufficient to address the multi-objective problem. This is feasible because different objectives are related to each other in many applications, and historical data and expert knowledge can provide this greater value.
>
>
> ----
>
> **Q7: However, a more detailed description of real-life application is appreciated. I surmise that this may help me to understand why we can assume that $\lambda$ is known.**
>
> We first provide a more detailed discription of two real-world applications metioned in our paper. One practical application is radiation treatment for patients, where the primary objective is target coverage and the secondary objective is the proximity of therapy to organs at risk (Jee et al., 2007). Another one is water resource planning for optimizing flood protection, supply shortage for irrigation, and electricity generation (Weber et al., 2002), where the water authorities legally mandate the prioritization of these three objectives and release policies over various time horizons.
>
> Based on the discussion in \textbf{Q6}, we only need to obtain an upper bound for the ratio of the change rates of different objectives. In above two applications, historical data and expert knowledge can provide this upper bound.
>
> ----
>
> **Q8: In the example given in the first paragraph, what does 'click-conversion rate' mean?**
>
> The click-conversion rate (CVR) refers to the percentage of visitors who perform a specific action on a website compared to the total number of visitors. For example,  CVR of e-commerce websites is the proportion of people who click and make purchases out of the total number of clicks.

---

> ### Author Response · Authors · 2023-11-17
> **Response to Reviewer vLB6 (Part III)**
>
> **Q9: Moreover, experiments with real-world data may make the efficiency of proposed algorithms more convincing.**
>
> Thank you for raising this issue. Following the experiments of [1], we will do more experiments by incorporating the Yahoo! Webscope dataset R6A in the final version. R6A is a real-world dataset collected from a personalized news recommender system, whose first objective is the click-through rate (CTR) and the second is the average payment from advertisers.
>
> *[1] Cem Tekin and Eralp Turgay. Multi-objective contextual multi-armed bandit with a dominant objective. IEEE Transactions on Signal Processing, 66(14):3799–3813, 2018.*
>
> ----
>
> **Q10: Weaknesses on presentation.**
>
> We appreciate your constructive reviews. We will incorporate two tables for comparing existing works and presenting the notations in the final version of our paper.
>
> ----
>
> **Q11: In the second paragraph of the introduction, it states that 'Therefore, if the evaluation criterion is Pareto regret, the agent can select any of the m objectives to optimize and ignore other objectives, which is unreasonable.' This statement is indeed strong. The author(s) may consider to explain this point a little bit more.**
>
> Theorem~$4.1$ of Xu $\\&$ Klabjan (2023) states that Pareto regret is smaller than the regret of any objective $i\in[m]$. Thus, we can achieve the nearly optimal Pareto regret bound by applying the UCB strategy to only one of the $m$ objectives. However, the remaining $m-1$ objectives still suffer linear regret bounds $O(T)$.
>
> Thank you for your constructive reviews. We will make a more detailed discussion about this point in the final version.
>
> ----
>
> **Q12: Minor suggestion in the second paragraph of Section 3.1.:  'We provide a formal definition of the chain relation to facilitate our presentation' may be a better expression than 'We give a formal definition of the chain relation to facilitate our presentation'.**
>
> Thank you for your detailed reviews. We will revise our paper according to your suggestions.

---

> ### Comment · Reviewer_vLB6 · 2023-11-22
>
> Thanks for your detailed response. I just increased the score to 5.
>
> I also went through comments from other reviewers. I may further review my score according to further responses from other reviewers.

---

> > ### Author Response · Authors · 2023-11-22
> > **Many thanks!**
> >
> > Dear Reviewer vLB6,
> >
> > Thank you very much for your kind reply! We are happy to answer more questions and sincerely hope the reviewer could reevaluate our paper.
> >
> > Best
> > Authors

---

### Official Review · Reviewer_1R9s · 2023-11-01

**Soundness:** 1 poor
**Presentation:** 2 fair
**Contribution:** 2 fair
**Rating:** 3
**Confidence:** 4

**Summary:**

This paper considers minimizing the regret with respect to the set of optimal arms defined according to the lexicographic ordering in stochastic linear bandits. It extends the prior work by incorporating linearly parameterized expected arm rewards. Compared to the prior work, it considers a relaxed notion of regret and shows that improved algorithms with regrets on the order of $\tilde{O}(\sqrt{T})$ can be designed. Their upper bounds diverge from classic stochastic linear bandits and multi-objective bandits in the sense that they involve a problem-dependent parameter $\lambda$ related to the rate at which different objective values change.

**Strengths:**

1) The study of multi-objective bandits under lexicographic ordering is a relatively under-explored area in the bandit literature. To the best of my knowledge, this paper is the first to analyze stochastic linear bandits under this setting. There exist several real-world applications where lexicographic ordering is important, making this topic worthy of investigation.


2) Introduction of $\lambda$ to capture the complexity of the optimal arm and an analysis based on that are the novel parts of this work. This allows the use of a new lexicographically ordered arm filter instead of the chain relation proposed in the prior work and, under the assumptions of the current work, yields new regret bounds.

**Weaknesses:**

1) It is not true that $R^i(T)$ is more stringent than $\hat{R}^i(T)$. For instance, let $\theta^1=(1,0)$, $\theta^2=(0,1)$, $x_A=(0,0)$, $x_B=(-1,1)$. In this case, $x_A$ is the lexicographic optimal arm. For a policy that always plays arm $x_B$, the $T$ round regrets are $R^1=T$, $R^2=-T$ and $\hat{R}^1=T$, $\hat{R}^2=0$. For his policy, the priority-based regret is greater than the general regret. It is also the case that priority-based regret should be harder to optimize than general regret since for the former, the lower bound is $\Omega(T^{2/3})$, while for the latter, this paper shows upper bounds of $\tilde{O}(T^{1/2})$ (but with dependence on $\lambda$ which seems to be problem-dependent).

2) Huyuk & Tekin (2021) also consider MOMAB under lexicographic ordering. They consider both priority-based regret, which is given in Equation 1, and general regret, which is given in Equation 4 (they call it priority-free regret). They have a lower bound on the priority-based regret, which is $\Omega(T^{2/3})$. This shows that MOMAB under lexicographic ordering with priority-based regret is harder than single-objective MAB for which $\tilde{O}(T^{1/2})$ upper bounds are possible. Therefore, the claim in the introduction saying that “prior work has proposed a suboptimal algorithm since the optimal regret bound for the existing single objective MAB algorithms is $O(K \log T)$” is incorrect.

3) From an algorithmic point of view, this paper provides novel techniques. However, the discussion with the prior work confuses the reader. There are some notable differences between the prior work and this work. The authors briefly mention that their $\tilde{O}(\sqrt{T})$ high-probability upper bounds do not contradict the $\Omega(T^{2/3})$ lower bound on the expected regret in the remark after Theorem 2. The authors justify this by saying that they focus on the pseudo-regret instead of the expected regret. I think that something is missing from the key comparison with the prior work. The proposed regret bounds are not instance-independent as they are given in terms of $\lambda$, an important parameter that represents the tradeoffs between different objectives. Utilizing this prior knowledge is crucial in achieving the time order of $\tilde{O}(\sqrt{T})$. One would expect that if the problem instance is adversarially chosen, then the algorithm proposed in the current work will not be able to achieve the claimed $\tilde{O}(\sqrt{T})$ upper bounds. For instance, given a bound on the $l_2$ norm of the true parameter vectors and $T$-round decision sets, can’t an adversary choose the true parameter vectors and decision sets such that lambda becomes a function of $T$? For the worst choice of lambda what will be the upper bounds?

4) Related to the above question, consider the same $\theta$ values in my first example, and let $x_A = (1+\epsilon, 0)$, $\epsilon>0$ and $x_B = (1,1)$. Clearly, $x_A$ is the lexicographic optimal arm. For these arms, Equation 5 in the paper gives $1 \leq \lambda \epsilon$. Thus $\lambda \geq 1/\epsilon$. Let $\epsilon = T^{-1/3}$. According to Theorem 2, $R^2(T)$ of MTE$^2$LO becomes $\tilde{O}(T^{2/3})$. One should be able to construct similar examples for the case with more than two objectives. When the number of objectives increases, I expect the time order of the regret upper bound to get even worse since $i$th objective’s time order depends on $\lambda^{i-1}$.

5) Based on the above discussion, the justification that the regret bounds are order-optimal because the scalar linear contextual bandit has a regret lower bound of $\Omega(d \sqrt{T})$ is unclear. Nearly matching upper bounds of algorithms such as LinUCB are minimax. Other than time and dimension of the feature vectors, the bounds only depend on upper bounds on the $l_2$ norms of the parameter and action vectors. It is unclear if the claim at Equation 5 is satisfied under the mild assumptions required to derive the $\tilde{O} (d \sqrt{T})$ for the scalar case (which seems not possible, based on the counter-example above).

**Questions:**

Please respond to the points mentioned in the weaknesses section.

---

> ### Author Response · Authors · 2023-11-17
> **Response to Reviewer 1R9s (Part I)**
>
> Many thanks for your constructive reviews. We have carefully considered your concerns and our responses are provided as follows.
>
> ----
>
> **Q1: It is not true that $R^i(T)$ is more stringent than $\hat{R}^i(T)$.**
>
> According to the common definition of regret (Auer, 2002; Abbasi-yadkori et al., 2011), regret is used to measure the cumulative gap between the selected arm and the best arm. However, the existing priority-based regret $\hat{R}^i(T)$ does not consider the instantaneous regret with false indicator function. In the infinite-armed setting, if the agent chooses the arms that approach the optimal arm at a rate of $1/\sqrt{T}$ in the first objective (the agent does not choose the optimal arm), then the regret for the first objective is $O(\sqrt{T})$, while all the remaining $m-1$ objectives have a regret of $0$ when measured by priority-based regret, which is clearly unreasonable.
>
> The reviewer provided a clever example to demonstrate that our expression "more stringent" is not concise. In the final version of our paper, we will revise it as "$R^i(T)$ accurately measures the performance of the agent compared to $\hat{R}^i(T)$".
>
> ----
>
> **Q2: Huyuk $\\&$ Tekin (2021) also consider MOMAB under lexicographic ordering ... This shows that MOMAB under lexicographic ordering with priority-based regret is harder than single-objective MAB for which upper bounds $\tilde{O}(T^{1/2})$ are possible. Therefore, the claim in the introduction saying that “prior work has proposed a suboptimal algorithm since the optimal regret bound for the existing single objective MAB algorithms is $O(K \log T)$” is incorrect.**
>
> The lower bound $\Omega(T^{2/3})$ is expected regret, while the upper bound $\tilde{O}(T^{1/2})$ is pseudo regret. Expected regret is greater than pseudo regret according to their definitions (Lattimore $\\&$ Szepesvari, 2020). Additionally, the lower bound of Huyuk $\\&$ Tekin (2021) is constructed based on the priority-based regret, while the upper bound is a general regret bound. Therefore, we cannot claim that "MOMAB under lexicographic ordering with priority-based regret is harder than single-objective MAB" based on a comparison of the expected lower bound and pseudo upper regret bound.
>
> The upper bound proposed by Huyuk $\\&$ Tekin (2021) is $\tilde{O}((KT)^{2/3})$, which is inconsistent with the optimal single objective MAB algorithm that has a regret bound of $O(K\log T)$. Even if the algorithm of Huyuk $\\&$ Tekin (2021) is applied to a single objective bandit problem, it still holds the regret bound $\tilde{O}((KT)^{2/3})$. In our paper, we propose the multiobjective bandit algorithm MTE$^2$LO, which achieves a regret bound of $\tilde{O}(\lambda^{i-1}d\sqrt{T})$ for the $i$-th objective. This bound is consistent with the optimal single objective bandit algorithm in terms of $d$ and $T$. Another advantage of MTE$^2$LO is that that if MTE$^2$LO is applied to a single objective SLB problem, it can achieve the same regret bound $\tilde{O}(d\sqrt{T})$ as the optimal single objective SLB algorithm.
>
> ----
>
> **Q3: From an algorithmic point of view, this paper provides novel techniques ... The proposed regret bounds are not instance-independent as they are given in terms of $\lambda$, an important parameter that represents the tradeoffs between different objectives.**
>
> Extending the bandit problem from a single objective to multiple objectives significantly increases its complexity, as the reward space shifts from one-dimensional to high-dimensional. In order to identify the optimal arm, the multiobjective algorithm has to search for the highest expected reward in the high-dimensional space. Consequently,  multiobjective bandit problem needs to take on a new form of regret bound that incorporates a new parameter. The parameter can be used to describe the complexity of the multiobjective problem itself. We propose the parameter $\lambda$. In the following, we take MAB and SLB as examples to support the rationality of adopting this parameter.
>
> The single objective MAB problem describes its complexity through the number of arms $K$ and the gap between suboptimal expected rewards and the optimal reward (Auer, 2002). The single objective SLB problem describes its complexity through the dimension of inherent parameter $d$ (Abbasi-yadkori et al., 2011). Our paper introduces the parameter $\lambda$ to capture the complexity of dealing with the multi-objective bandit problem under lexicographic ordering. Furthermore, the regret bound for the first objective is $\tilde{O}(d\sqrt{T})$, which aligns with the optimal single objective bandit algorithm. This provides support for the claim that the introduction of $\lambda$ in the regret bound is a result of the reward space expanding from one-dimensional to high-dimensional.

---

> ### Author Response · Authors · 2023-11-17
> **Response to Reviewer 1R9s (Part II)**
>
> **Q4: One would expect that if the problem instance is adversarially chosen, then the algorithm proposed in the current work will not be able to achieve the claimed $\tilde{O}(\sqrt{T})$ upper bound.**
>
> As we have discussed in **Q3**, $\lambda$ in the multi-objective bandit problem is a instance-dependent parameter similar to the number of arms $K$ in MAB and the feature dimension $d$ in SLB. To illustrate the adversarially chosen issue, we use the single objective SLB as an example. The upper regret bound of the single objective SLB is $\tilde{O}(d\sqrt{T})$. An adversary can select a decision set where $d$ becomes a function of $T$, resulting in a higher order dependence on $T$ rather than just $\sqrt{T}$. However, we cannot conclude that the upper regret bound of the single objective SLB is not $\tilde{O}(\sqrt{T})$.
>
> The parameter $\lambda$ captures the complexity of identifying the lexicographic optimal arm, and it does not depend on $T$ for a fixed bandit problem. Therefore, $\tilde{O}(\lambda^{i-1}d\sqrt{T}/T)$ tends to zero at a rate of $\tilde{O}(1/\sqrt{T})$ as $T$ approaches infinity, which is the same rate as the single objective algorithm.
>
> ----
>
> **Q5: Based on the above discussion, the justification that the regret bounds are order-optimal because the scalar linear contextual bandit has a regret lower bound of $\Omega(d\sqrt{T})$ is unclear. Nearly matching upper bounds of algorithms such as LinUCB are minimax. Other than time and dimension of the feature vectors, the bounds only depend on upper bounds on the $\ell_2$ norms of the parameter and action vectors.**
>
> As we have discussed in **Q3** and **Q4**, when extending the reward space from one-dimensional to high-dimensional, it becomes necessary to introduce a new parameter that can measure the complexity of the high-dimensional problem. We have introduced a novel parameter $\lambda$, developed an effective algorithm MTE$^2$LO, and provided corresponding theoretical guarantees for its performance.
>
> For the multiobjective bandit problem under lexicographic ordering, it may be more appropriate to consider a regret bound of the form $O(dT^{f(i)})$ for the $i$-th objective, where $f(\cdot)$ monotonically increases with respect to the objective $i$ and $f(1)=1/2$. However, we have no idea how to obtain this form of regret bound. The regret bound presented in our paper aligns with that of the single objective bandit problem, thereby offering valuable insights into the multiobjective bandit problem under lexicographic ordering.
>
> The reviews offer much insights to our research. We are happy to answer more questions and sincerely hope the reviewer could reevaluate our paper.

---

> ### Author Response · Authors · 2023-11-22
> **Many thanks for your review and we are open for more questions.**
>
> Dear Reviewer 1R9s,
>
> Thank you again for your time and efforts in reviewing our paper. We hope that our response addresses your concerns and questions.
>
> As our author-reviewer discussion nears its end, we'd appreciate knowing if your concerns are resolved.  We are happy to answer more questions and would appreciate a reassessment of our paper.
>
> Best
> Authors

---

### Official Review · Reviewer_1nFM · 2023-11-01

**Soundness:** 2 fair
**Presentation:** 2 fair
**Contribution:** 2 fair
**Rating:** 3
**Confidence:** 4

**Summary:**

The paper introduces the multi-objective stochastic linear bandit model under lexicographic ordering, where in the priority ordering is given by the indices of the objectives. The work claims to achieve close to optimal regret bounds with the general regret metric.The novelty of the paper lies in the new arm filtering algorithm $LOAF$ and multiple trade off approach for exploration and exploitation in $MTE^2LO$.

**Strengths:**

The paper gives a clear and significant background on MAB and MOMAB before introducing the main algorithms, leading to ease of understanding.

**Weaknesses:**

1. Paper fails to reasonably motivate why a simple algorithm such as mentioned below would not do better -
If we have the lexicographic ordering as defined in the paper then why not apply a simple OFUL just for 1st objective and if there are multiple arms at the end of 1st OFUL then apply OFUL for second objective and so on? Would this not achieve similar to abbasi regret?

2. Further the theorem's mentioned in the paper do not talk about the finiteness or infinite decision set $\mathcal{D}$, if the decision set is finite then the algorithm needs to be compared with that of SupLinUCB regret bounds, and if the decision set is infinite the how is it even possible to filter the arms based on chain relations?

3. I fail to understand how is the new regret formulation different from the previous regret formulation with indicator? Because if the indicator function is false then that would inherently increase the regret of the objective for different $i$.

4. In the paragraph before equation 4, it's mentioned that the decision set for all the times are determined before the game start how is it possible?

5. Having a prior knowledge of parameter $\lambda$ seems infeasible.

6. In section 2.2, it's mentioned that Lu et al., achieves Pareto regret bound that is optimal, doesn't Pareto optimal regret imply lexicographical optimal regret? How does this work go beyond this prior work?

7. Finally, there is little to no intuition on why there are 3 different phases in $MTE^2LO$?

**Questions:**

Questions mentioned in the above section and on Page 5 what is $SCE^2LO$ is it a typo for $STE^2LO$?

---

> ### Author Response · Authors · 2023-11-17
> **Response to Reviewer 1nFM (Part I)**
>
> Many thanks for your constructive reviews. We have carefully considered your concerns and our responses are provided as follows.
>
> ----
>
> **Q1: Paper fails to reasonably motivate why a simple algorithm such as mentioned below would not do better ... Would this not achieve similar to abbasi regret?**
>
> OFUL selects the arm with the highest upper confidence bound. For the 1st objective, OFUL cannot guarantee that all arms with the highest expected rewards have the same highest upper confidence bound due to the uncertainty in the upper confidence bound. As a result, OFUL may only output a **single** arm and fail to consider the remaining $m-1$ objectives.
>
> ----
>
> **Q2: Further the theorem's mentioned in the paper do not talk about the finiteness or infinite decision set $\mathcal{D}$ ... how is it even possible to filter the arms based on chain relations?**
>
> We present two algorithms to deal with the multi-objective bandits under lexicographic ordering. The first algorithm, STE$^2$LO, can only handle a finite decision set ($|\mathcal{D}_t|=K$), as we stated in Theorem 1. STE$^2$LO filters the arms by the chain relation. The second algorithm, MTE$^2$LO, is designed to handle the infinite decision set case. We design a novel filter called LOAF, which utilizes the intersection of scaling confidence intervals to filter the infinite arms. More details about the motivation for LOAF is explained at the beginning of Section 3.2.
>
> ----
>
> **Q3: I fail to understand how is the new regret formulation different from the previous regret formulation with indicator?**
>
> The priority-based regret $\widehat{R}^i(T)$ of Hüyük $\\&$ Tekin (2021) is defined as follows:
> $$
> \widehat{R}\^i(T)=\sum\_{t=1}\^T\langle \theta\_\*\^i, x\_t\^\* - x\_t\rangle\mathbb{I}(\langle \theta\_\*\^j, x\_t\^\*\rangle = \langle \theta\_\*\^j, x\_t\rangle, 1\leq j\leq i-1), i=1,2,\dots,m
> $$
> where $x_t^\*$ denotes the optimal arm in $\mathcal{D}\_t$ according to the lexicographic order. Note that the priority-based regret $\widehat{R}^i(T)$ relies on the indicator function $\mathbb{I}(\cdot)$, which cannot accumulate the instantaneous gap $\langle \theta\_\*^i, x\_t^\* - x\_t\rangle$ for $i\geq 2$ if $\langle \theta_\*^1, x\_t\rangle < \langle \theta\_\*^1, x\_t^\*\rangle$. Thus, $\widehat{R}^i(T)$ can not accurately measure the performance of agent for $i\geq2$.
>
> Our proposed regret $R^i(T)$ is a commonly used metric in the single-objective bandit field (Auer, 2002; Bubeck $\\&$ Cesa-Bianchi, 2012; Lattimore $\\&$ Szepesvári, 2020), and we simply extend it to $m$ objectives, such that
> $$
> R^i(T)=\sum\_{t=1}\^T\langle \theta_\*^i, x_t^\* - x_t\rangle, i=1,2,\dots,m.
> $$
> $R^i(T)$ removes the indicator function, enabling the accumulation of the rewards gap between the selected arm $x_t$ and the lexicographically optimal arm $x_t^*$ independently for all objectives $i=1,2,\ldots,m$. This makes it a more reliable metric for reflecting the algorithms' performance.
>
> ----
>
> **Q4: In the paragraph before equation 4, it's mentioned that the decision set for all the times are determined before the game start how is it possible?**
>
> Here we want to ensure the arm sets do not depend on the decision of the learner, so the decision sets are fixed beforehand. This is the standard setting of stochastic linear bandits (Dani et al., 2008; Abbasi-yadkori et al., 2011). We will further explain this setting in the final version.
>
> ----
>
> **Q5: Having a prior knowledge of parameter $\lambda$ seems infeasible.**
>
> The dependence on $\lambda$ is indeed a limitation, as mentioned in the conclusion and future work section. However, in most multi-objective scenarios, it is possible to estimate $\lambda$. As we have discussed under Equation (5), $\lambda$ can be estimated by measuring how different objectives change as the decision varies. To further illustrate this, we provide a simple example with two objectives.
>
> The expected reward functions are denoted as $\mu^1(x)$ and $\mu^2(x)$, which is defined as $\mu^1(x)=1-\min_{p\in\\{0.4,0.8\\}}|x-p|$ and $\mu^2(x)=1-2|x-0.3|$ (Here we do not take a linear function so as to highlight the change rates of  expected reward functions). The optimal arms for the first objective are $\\{0.4, 0.8\\}$, and the lexicographic optimal arm is $0.4$. In this example, the value of $\lambda$ is 2, which is the ratio of the change rates of $\mu^1(x)$ and $\mu^2(x)$.
>
> Another point worth mentioning is that it is not necessary to know the **precise** ratio of the change rates. Any value **greater** than the ratio of the change rates is sufficient to address the multi-objective problem. This is feasible because different objectives are related to each other in many applications, and historical data and expert knowledge can provide this greater value.

---

> ### Author Response · Authors · 2023-11-17
> **Response to Reviewer 1nFM (Part II)**
>
> **Q6: In section 2.2, it's mentioned that Lu et al., achieves Pareto regret bound that is optimal, doesn't Pareto optimal regret imply lexicographical optimal regret? How does this work go beyond this prior work?**
>
> Theorem~$4.1$ of Xu $\\&$ Klabjan (2023) states that Pareto regret is smaller than the regret of any objective $i\in[m]$. Thus, we can achieve the nearly optimal Pareto regret bound by applying the UCB strategy to only one of the $m$ objectives. However, the remaining $m-1$ objectives still suffer linear regret bounds $O(T)$. Our proposed algorithm MTE$^2$LO achieves the nearly optimal regret bounds $\tilde{O}(d\sqrt{T})$ for all objectives simultaneously. The key innovations of our algorithm include the introduction of a new arm filter called LOAF and the implementation of a multi-stage decision-making strategy (Step 4 to Step 15 of MTE$^2$LO).
>
> ----
>
> **Q7: Finally, there is little to no intuition on why there are 3 different phases in MTE$^2$LO?**
>
> One important reason for dividing the decision-making operation into three parts is that the LOAF algorithm requires the width of the confidence interval for input arms to be no greater than $W$. MTE$^2$LO utilizes LOAF as a subroutine to filter arms. Therefore, MTE$^2$LO divides the decision-making operation into three parts in order to incorporate LOAF in the bounded confidence interval cases ("if" and "else" cases). To further strike a balance between exploration and exploitation, the decision-making process in each round is divided into $S$ stages, enabling the achievement of the nearly optimal regret bound $\tilde{O}(d\sqrt{T})$.
>
> ----
>
> **Q8: What is SCE$^2$LO? Is it a typo for STE$^2$LO?**
>
> Thank you for raising this issue. We will make a detailed revise to avoid typos in the final version.

---

> ### Author Response · Authors · 2023-11-22
> **Many thanks for your review and we are open for more questions.**
>
> Dear Reviewer 1nFM,
>
> Thank you again for your time and efforts in reviewing our paper. We hope that our response addresses your concerns and questions.
>
> As our author-reviewer discussion nears its end, we'd appreciate knowing if your concerns are resolved.  We are happy to answer more questions and would appreciate a reassessment of our paper.
>
> Best
> Authors

---

> ### Comment · Reviewer_1nFM · 2023-11-22
>
> Dear Authors,
>
> Thank you for the detailed response. I have also gone through the comments of other reviewers as well. I have decided to keep my score.

---

> > ### Author Response · Authors · 2023-11-22
> >
> > Dear Reviewer 1nFM,
> >
> > Thank you for your constructive reviews. We'd appreciate knowing what your further concerns are. We are happy to answer more questions and collaborate to enhance the quality of our paper.
> >
> > Best
> > Authors

---

### Official Review · Reviewer_chLG · 2023-11-04

**Soundness:** 3 good
**Presentation:** 4 excellent
**Contribution:** 2 fair
**Rating:** 5
**Confidence:** 3

**Summary:**

The paper tackles the multiobjective multi-armed bandit (MOMAB) problem with objective ordering. Specifically, the paper uses lexicographic ordering to solve the issue of priority in the MOMAB setup.

The paper proposes STE$^2$LO algorithm, an improvement of already established PF-LEX, and MTE$^2$LO algorithms which use lexicographic ordering to solve MOMAB and prove regret bound for the same. MTE$^2$LO is proved to have a regret upper bound of $\mathcal{O}(d\sqrt{T})$.

Experimental evidence shows the prowess of the proposed algorithms.

**Strengths:**

The paper tackles an interesting and important problem of integrating priority of objectives in Multi-armed bandit algorithms. The method proposed is through the utilization of lexicographic ordering.

The paper provides sufficient explanation and intuition on the novel and interesting improvements from PF-LEX to STE$^2$LO to MTE$^2$LO.

The regret upper bounds are proved and stated clearly for both the proposed algorithms and experimental evidence is provided.

Overall the paper is clearly written in terms of mathematical notation, definitions, assumptions, algorithms, and proofs.

**Weaknesses:**

Three central concerns are highlighted as follows:

1. **Applying single objective MAB algorithms multiple times**: Would the authors provide arguments as to why it wouldn't be a good idea to just use a standard single objective MAB routine multiple times based on the objective priority order and solve the MOMAB setup? Even with a naive bound, regret of $\mathcal{O}(md\sqrt{T})$ would be achievable. This compared to the current regret bound in this paper $\mathcal{O}((\sum_{i}\lambda^i)d\sqrt{T})$ isn't clear to be a guaranteed improvement at first glance. I am willing to be completely wrong about this point.

2. **Intuition and necessity of $\lambda^i$**: Firstly, the paper keeps referencing $\lambda$ when technically it is $\lambda^i$ for each objective (please correct me if this is not the case). Secondly, $\lambda^i$ serves to establish some sort of regularity between the different objectives. I fail to see the need or necessity of doing so. If this is utterly needed, can you provide a counter-example of things going completely haywire in the absence of such regularity condition?

Personally, I am just confused about the need for such a regularity. Hope the authors can shed some light on this.

3. **Simulations**: Is it possible to perform a simulation on a real-world dataset? Another query is that the plot appears too linear for all the algorithms even with 10000 steps. Can you either run the algorithms for longer or showcase a smaller setup where the sublinear part of the curve is visible?

While these are the major concerns. I am completely ready to be proven wrong and will gladly increase my score based on the author's comments.

**Questions:**

Is forced exploration required, can we not employ UCB-like schemes that take care of exploration-exploitation inherently? Why was this
three-part exploration-exploitation preferred over UCB like universal choice?

---

> ### Author Response · Authors · 2023-11-17
>
> Many thanks for your constructive reviews. We have carefully considered your concerns and our responses are provided as follows.
>
> ----
>
> **Q1: Would the authors provide arguments as to why it wouldn't be a good idea to just use a standard single objective MAB routine multiple times based on the objective priority order and solve the MOMAB setup?**
>
> Applying a standard single objective MAB routine multiple times can only achieve a linear regret bound of $O(T)$. To illustrate this issue, we consider a 2-objective problem. Specifically, when applying a standard UCB strategy to the first objective, it outputs a **single** arm $\hat{x}_t^1$ that is promising for the first objective. However, the second objective is not taken into account at all, resulting in a linear regret bound $O(T)$ for the second objective. Therefore, we design a mechanism that can filter **a group of** arms $D_t^1$ from $D_t$, where $D_t^1$ are promising for the first objective. Then, we select the most promising arm for the second objective from $D_t^1$, which can guarantee sublinear regret bounds for both the first objective and second objective. The detailed procedure of our designed filter mechanism is presented in Algorithm 2, and its theoretical guarantee is provided by Proposition 1.
>
> ----
>
> **Q2: Firstly, the paper keeps referencing $\lambda$ ...  I fail to see the need or necessity of doing so. If this is utterly needed, can you provide a counter-example of things going completely haywire in the absence of such regularity condition?**
>
> Our first algorithm, called STE$^2$LO, does not utilize $\lambda$ and can only achieve a regret bound of $\tilde{O}(T^{2/3})$, which is far from the nearly optimal regret bound of $\tilde{O}(\sqrt{T})$. In contrast, our second algorithm, MTE$^2$LO, incorporates the parameter $\lambda$ and improves the regret bound to $\tilde{O}(\sqrt{T})$. The main purpose of $\lambda$ is to scale the confidence interval, which prevents the absence of the optimal arm during the filtering process in the algorithm LOAF. For more details on the motivation behind adopting $\lambda$, please refer to the beginning of Section 3.2.
>
> ----
>
> **Q3: Simulations: Is it possible to perform a simulation on a real-world dataset?**
>
> Following the experiments of [1], we will do more experiments by incorporating the Yahoo! Webscope dataset R6A in the final version. R6A is a real-world dataset collected from a personalized news recommender system, whose first objective is the click-through rate (CTR), and the second is the average payment from advertisers.
>
> *[1] Cem Tekin and Eralp Turgay. Multi-objective contextual multi-armed bandit with a dominant objective. IEEE Transactions on Signal Processing, 66(14):3799–3813, 2018.*
>
> ----
>
> **Q4: Simulations: Can you either run the algorithms for longer or showcase a smaller setup where the sublinear part of the curve is visible?**
>
> Thank you for raising this issue. We will run the algorithms with $T=10^6$ and fine-tune these algorithms to demonstrate the sublinear convergence in the final version of our paper.
>
> ----
>
> **Q5: Is forced exploration required, can we not employ UCB-like schemes that take care of exploration-exploitation inherently? Why was this three-part exploration-exploitation preferred over UCB like universal choice?**
>
> To our knowledge, UCB-like strategies cannot solve the problems studied in our paper. One important reason for dividing the decision-making operation into three parts is that the LOAF algorithm requires the width of the confidence interval for input arms to be no greater than $W$. MTE$^2$LO utilizes LOAF as a subroutine to filter arms. Therefore, MTE$^2$LO divides the decision-making operation into three parts in order to incorporate LOAF in the bounded confidence interval cases ("if" and "else" cases). To further strike a balance between exploration and exploitation, the decision-making process in each round is divided into $S$ stages, enabling the achievement of the nearly optimal regret bound $\tilde{O}(d\sqrt{T})$.

---

> ### Author Response · Authors · 2023-11-22
> **Many thanks for your review and we are open for more questions.**
>
> Dear Reviewer chLG,
>
> Thank you again for your time and efforts in reviewing our paper. We hope that our response addresses your concerns and questions.
>
> As our author-reviewer discussion nears its end, we'd appreciate knowing if your concerns are resolved.  We are happy to answer more questions and would appreciate a reassessment of our paper.
>
> Best
> Authors

---

### Meta-Review · Area_Chair_Sd6P · 2023-12-04

**Metareview:**

This paper looks at a multi-objective bandit problem where the relative "importance" of the objectives is settled via the lexicographic ordering.

The reviews are lukewarm, hence I looked at the paper myself. I agree that the setting is interesting, not brand new, but the major downside is indeed the prior knowledge of a key problem parameter (some lambda). This is a very strong assumption to me, that prevents this paper to reach the ICLR bar.

**Justification For Why Not Higher Score:**

This paper does not reach the bar.

**Justification For Why Not Lower Score:**

N/A

---

### Decision · Program_Chairs · 2024-01-16

Reject